# TRANSFORMER-BASED WORLD MODELS ARE HAPPY WITH 100K INTERACTIONS

**Jan Robine, Marc Höftmann, Tobias Uelwer, Stefan Harmeling**
Department of Computer Science, Technical University of Dortmund, Germany

## ABSTRACT

Deep neural networks have been successful in many reinforcement learning settings. However, compared to human learners they are overly data hungry. To build a sample-efficient world model, we apply a transformer to real-world episodes in an autoregressive manner: not only the compact latent states and the taken actions but also the experienced or predicted rewards are fed into the transformer, so that it can attend flexibly to all three modalities at different time steps. The transformer allows our world model to access previous states directly, instead of viewing them through a compressed recurrent state. By utilizing the Transformer-XL architecture, it is able to learn long-term dependencies while staying computationally efficient. Our transformer-based world model (TWM) generates meaningful, new experience, which is used to train a policy that outperforms previous model-free and model-based reinforcement learning algorithms on the Atari 100k benchmark. Our code is available at `https://github.com/jrobine/twm`.

## 1 INTRODUCTION

Deep reinforcement learning methods have shown great success on many challenging decision making problems. Notable methods include DQN (Mnih et al., 2015), PPO (Schulman et al., 2017), and MuZero (Schrittwieser et al., 2019). However, most algorithms require hundreds of millions of interactions with the environment, whereas humans often can achieve similar results with less than 1% of these interactions, i.e., they are more sample-efficient. The large amount of data that is necessary renders a lot of potential real world applications of reinforcement learning impossible.

Recent works have made a lot of progress in advancing the sample efficiency of RL algorithms: model-free methods have been improved with auxiliary objectives (Laskin et al., 2020b), data augmentation (Yarats et al., 2021, Laskin et al., 2020a), or both (Schwarzer et al., 2021). Model-based methods have been successfully applied to complex image-based environments and have either been used for planning, such as EfficientZero (Ye et al., 2021), or for learning behaviors in imagination, such as SimPLe (Kaiser et al., 2020).

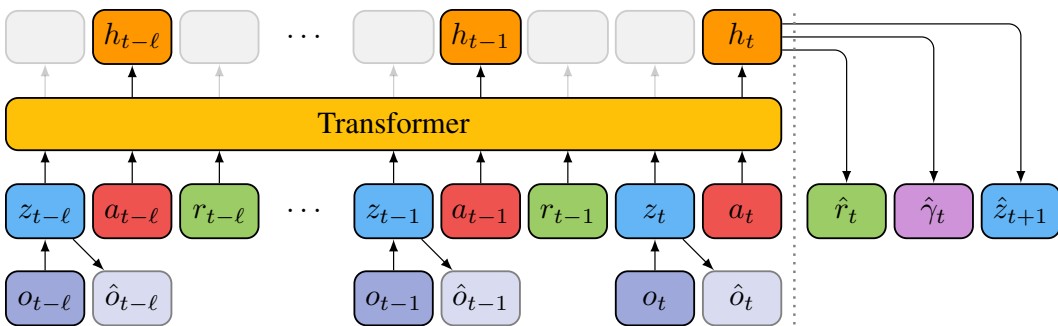

Figure 1: Our world model architecture. Observations $o_{t-\ell:t}$ are encoded using a CNN. Linear embeddings of stochastic, discrete latent states $z_{t-\ell:t}$, actions $a_{t-\ell:t}$, and rewards $r_{t-\ell:t}$ are fed into a transformer, which computes a deterministic hidden state $h_t$ at each time step. Predictions of the reward $r_t$, discount factor $\gamma_t$, and next latent state $z_{t+1}$ are computed based on $h_t$ using MLPs.

A promising model-based concept is learning in imagination (Ha & Schmidhuber, 2018; Kaiser et al., 2020; Hafner et al., 2020; Hafner et al., 2021): instead of learning behaviors from the collected experience directly, a generative model of the environment dynamics is learned in a (self-)supervised manner. Such a so-called *world model* can create new trajectories by iteratively predicting the next state and reward. This allows for potentially indefinite training data for the reinforcement learning algorithm without further interaction with the real environment. A world model might be able to generalize to new, unseen situations, because of the nature of deep neural networks, which has the potential to drastically increase the sample efficiency. This can be illustrated by a simple example: in the game of Pong, the paddles and the ball move independently. In the best case, a successfully trained world model would imagine trajectories with paddle and ball configurations that have never been observed before, which enables learning of improved behaviors.

In this paper, we propose to model the world with transformers (Vaswani et al., 2017), which have significantly advanced the field of natural language processing and have been successfully applied to computer vision tasks (Dosovitskiy et al., 2021). A transformer is a sequence model consisting of multiple self-attention layers with residual connections. In each self-attention layer the inputs are mapped to keys, queries, and values. The outputs are computed by weighting the values by the similarity of keys and queries. Combined with causal masking, which prevents the self-attention layers from accessing future time steps in the training sequence, transformers can be used as autoregressive generative models. The Transformer-XL architecture (Dai et al., 2019) is much more computationally efficient than vanilla transformers at inference time and introduces relative positional encodings, which remove the dependence on absolute time steps.

**Our contributions:**  The contributions of this work can be summarized as follows:

1. We present a new autoregressive world model based on the Transformer-XL (Dai et al., 2019) architecture and a model-free agent trained in latent imagination. Running our policy is computationally efficient, as the transformer is not needed at inference time. This is in contrast to related works (Hafner et al., 2020; 2021; Chen et al., 2022) that require the full world model during inference.

2. Our world model is provided with information on how much reward has already been emitted by feeding back predicted rewards into the world model. As shown in our ablation study, this improves performance.

3. We rewrite the balanced KL divergence loss of Hafner et al. (2021) to allow us to fine-tune the relative weight of the involved entropy and cross-entropy terms.

4. We introduce a new thresholded entropy loss that stabilizes the policy's entropy during training and hereby simplifies the selection of hyperparameters that behave well across different games.

5. We propose a new effective sampling procedure for the growing dataset of experience, which balances the training distribution to shift the focus towards the latest experience. We demonstrate the efficacy of this procedure with an ablation study.

6. We compare our transformer-based world model (TWM) on the Atari 100k benchmark with recent sample-efficient methods and obtain excellent results. Moreover, we report empirical confidence intervals of the aggregate metrics as suggested by Agarwal et al. (2021).

## 2  METHOD

We consider a partially observable Markov decision process (POMDP) with discrete time steps $t \in \mathbb{N}$, scalar rewards $r_t \in \mathbb{R}$, high-dimensional image observations $o_t \in \mathbb{R}^{h \times w \times c}$, and discrete actions $a_t \in \{1, \ldots, m\}$, which are generated by some policy $a_t \sim \pi(a_t \mid o_{1:t}, a_{1:t-1})$, where $o_{1:t}$ and $a_{1:t-1}$ denote the sequences of observations and actions up to time steps $t$ and $t - 1$, respectively. Episode ends are indicated by a boolean variable $d_t \in \{0, 1\}$. Observations, rewards, and episode ends are jointly generated by the unknown environment dynamics $o_t, r_t, d_t \sim p(o_t, r_t, d_t \mid o_{1:t-1}, a_{1:t-1})$. The goal is to find a policy $\pi$ that maximizes the expected sum of discounted rewards $\mathbb{E}_\pi\left[\sum_{t=1}^{\infty} \gamma^{t-1} r_t\right]$, where $\gamma \in [0, 1)$ is the discount factor. Learning in imagination consists of three steps that are repeated iteratively: learning the dynamics, learning a policy, and interacting in the real environment. In this section, we describe our world model and policy, concluding with the training procedure.

## 2.1 WORLD MODEL

Our world model consists of an observation model and a dynamics model, which do not share parameters. Figure 1 illustrates our combined world model architecture.

**Observation Model:** The observation model is a variational autoencoder (Kingma & Welling, 2014), which encodes observations $o_t$ into compact, stochastic latent states $z_t$ and reconstructs the observations with a decoder, which in our case is only required to obtain a learning signal for $z_t$:

$$\begin{aligned} \text{Observation encoder:} \quad & z_t \sim p_\phi(z_t \mid o_t) \\ \text{Observation decoder:} \quad & \hat{o}_t \sim p_\phi(\hat{o}_t \mid z_t). \end{aligned} \tag{1}$$

We adopt the neural network architecture of DreamerV2 (Hafner et al., 2021) with slight modifications for our observation model. Thus, a latent state $z_t$ is discrete and consists of a vector of 32 categorical variables with 32 categories. The observation decoder reconstructs the observation and predicts the means of independent standard normal distributions for all pixels. The role of the observation model is to capture only non-temporal information about the current time step, which is different from Hafner et al. (2021). However, we include short-time temporal information, since a single observation $o_t$ consists of four frames (aka frame stacking, see also Section 2.2).

**Autoregressive Dynamics Model:** The dynamics model predicts the next time step conditioned on the history of its past predictions. The backbone is a deterministic aggregation model $f_\psi$ which computes a deterministic hidden state $h_t$ based on the history of the $\ell$ previously generated latent states, actions, and rewards. Predictors for the reward, discount, and next latent state are conditioned on the hidden state. The dynamics model consists of these components:

$$\begin{aligned} \text{Aggregation model:} \quad & h_t = f_\psi(z_{t-\ell:t}, a_{t-\ell:t}, r_{t-\ell:t-1}) \\ \text{Reward predictor:} \quad & \hat{r}_t \sim p_\psi(\hat{r}_t \mid h_t) \\ \text{Discount predictor:} \quad & \hat{\gamma}_t \sim p_\psi(\hat{\gamma}_t \mid h_t) \\ \text{Latent state predictor:} \quad & \hat{z}_{t+1} \sim p_\psi(\hat{z}_{t+1} \mid h_t). \end{aligned} \tag{2}$$

The aggregation model is implemented as a causally masked Transformer-XL (Dai et al., 2019), which enhances vanilla transformers (Vaswani et al., 2017) with a recurrence mechanism and relative positional encodings. With these encodings, our world model learns the dynamics independent of absolute time steps. Following Chen et al. (2021), the latent states, actions, and rewards are sent into modality-specific linear embeddings before being passed to the transformer. The number of input tokens is $3\ell - 1$, because of the three modalities (latent states, actions, rewards) and the last reward not being part of the input. We consider the outputs of the action modality as the hidden states and disregard the outputs of the other two modalities (see Figure 1; orange boxes vs. gray boxes).

The latent state, reward, and discount predictors are implemented as multilayer perceptrons (MLPs) and compute the parameters of a vector of independent categorical distributions, a normal distribution, and a Bernoulli distribution, respectively, conditioned on the deterministic hidden state. The next state is determined by sampling from $p_\psi(\hat{z}_{t+1} \mid h_t)$. The reward and discount are determined by the mean of $p_\psi(\hat{r}_t \mid h_t)$ and $p_\psi(\hat{\gamma}_t \mid h_t)$, respectively.

As a consequence of these design choices, our world model has the following beneficial properties:

1. The dynamics model is autoregressive and has direct access to its previous outputs.
2. Training is efficient since sequences are processed in parallel (compared with RNNs).
3. Inference is efficient because outputs are cached (compared with vanilla Transformers).
4. Long-term dependencies can be captured by the recurrence mechanism.

We want to provide an intuition on why a fully autoregressive dynamics model is favorable: First, the direct access to previous latent states enables to model more complex dependencies between them, compared with RNNs, which only see them indirectly through a compressed recurrent state. This also has the potential to make inference more robust, since degenerate predictions can be ignored more easily. Second, because the model sees which rewards it has produced previously, it can react to its own predictions. This is even more significant when the rewards are sampled from a probability distribution, since the introduced noise cannot be observed without autoregression.

**Loss Functions:** The observation model can be interpreted as a variational autoencoder with a temporal prior, which is provided by the latent state predictor. The goal is to keep the distributions of the encoder and the latent state predictor close to each other, while slowly adapting to new observations and dynamics. Hafner et al. (2021) apply a balanced KL divergence loss, which lets them control which of the two distributions should be penalized more. To control the influences of its subterms more precisely, we disentangle this loss and obtain a *balanced cross-entropy loss* that computes the cross-entropy $H(p_\phi(z_{t+1} \mid o_{t+1}), p_\psi(\hat{z}_{t+1} \mid h_t))$ and the entropy $H(p_\phi(z_t \mid o_t))$ explicitly. Our derivation can be found in Appendix A.2. We call the cross-entropy term for the observation model the *consistency loss*, as its purpose is to prevent the encoder from diverging from the dynamics model. The entropy regularizes the latent states and prevents them from collapsing to one-hot distributions. The observation decoder is optimized via negative log-likelihood, which provides a rich learning signal for the latent states. In summary, we optimize a self-supervised loss function for the observation model that is the expected sum over the decoder loss, the entropy regularizer and the consistency loss

$$\mathcal{L}_\phi^{\text{Obs.}} = \mathbb{E}\left[ \sum_{t=1}^T \underbrace{-\ln p_\phi(o_t \mid z_t)}_{\text{decoder}} - \underbrace{\alpha_1 H(p_\phi(z_t \mid o_t))}_{\text{entropy regularizer}} + \underbrace{\alpha_2 H(p_\phi(z_t \mid o_t), p_\psi(\hat{z}_t \mid h_{t-1}))}_{\text{consistency}} \right], \quad (3)$$

where the hyperparameters $\alpha_1, \alpha_2 \geq 0$ control the relative weights of the terms.

For the balanced cross-entropy loss, we also minimize the cross-entropy in the loss of the dynamics model, which is how we train the latent state predictor. The reward and discount predictors are optimized via negative log-likelihood. This leads to a self-supervised loss for the dynamics model

$$\mathcal{L}_\psi^{\text{Dyn.}} = \mathbb{E}\left[ \sum_{t=1}^T \underbrace{H(p_\phi(z_{t+1} \mid o_{t+1}), p_\psi(\hat{z}_{t+1} \mid h_t))}_{\text{latent state predictor}} - \underbrace{\beta_1 \ln p_\psi(r_t \mid h_t)}_{\text{reward predictor}} - \underbrace{\beta_2 \ln p_\psi(\gamma_t \mid h_t)}_{\text{discount predictor}} \right], \quad (4)$$

with coefficients $\beta_1, \beta_2 \geq 0$ and where $\gamma_t = 0$ for episode ends ($d_t = 1$) and $\gamma_t = \gamma$ otherwise.

## 2.2 POLICY

Our policy $\pi_\theta(a_t \mid \hat{z}_t)$ is trained on imagined trajectories using a mainly standard advantage actor-critic (Mnih et al., 2016) approach. We train two separate networks: an actor $a_t \sim \pi_\theta(a_t \mid \hat{z}_t)$ with parameters $\theta$ and a critic $v_\xi(\hat{z}_t)$ with parameters $\xi$. We compute the advantages via Generalized Advantage Estimation (Schulman et al., 2016) while using the discount factors predicted by the world model $\hat{\gamma}_t$ instead of a fixed discount factor for all time steps. As in DreamerV2 (Hafner et al., 2021), we weight the losses of the actor and the critic by the cumulative product of the discount factors, in order to softly account for episode ends.

**Thresholded Entropy Loss:** We penalize the objective of the actor with a slightly modified version of the usual entropy regularization term (Mnih et al., 2016). Our penalty normalizes the entropy and only takes effect when the entropy falls below a certain threshold

$$\mathcal{L}_\theta^{\text{Ent.}} = \max\left( 0, \Gamma - \frac{H(\pi_\theta)}{\ln(m)} \right), \quad (5)$$

where $0 \leq \Gamma \leq 1$ is the threshold hyperparameter, $H(\pi_\theta)$ is the entropy of the policy, $m$ is the number of discrete actions, and $\ln(m)$ is the maximum possible entropy of the categorical action distribution. By doing this, we explicitly control the percentage of entropy that should be preserved across all games independent of the number of actions. This ensures exploration in the real environment and in imagination without the need for $\epsilon$-greedy action selection or changing the temperature of the action distribution. We also use the same stochastic policy when evaluating our agent in the experiments. The idea of applying a hinge loss to the entropy was first introduced by Pereyra et al. (2017) in the context of supervised learning. In Appendix A.1 we show the effect of this loss.

**Choice of Policy Input:** The policy computes an action distribution $\pi_\theta(a_t \mid x_t)$ given some view $x_t$ of the state. For instance, $x_t$ could be $o_t$, $z_t$, or $[z_t, h_t]$ at inference time, i.e., when applied to the real environment, or the respective predictions of the world model $\hat{o}_t$, $\hat{z}_t$, or $[\hat{z}_t, h_t]$ at training time. This view has to be chosen carefully, since it can have a significant impact on the performance

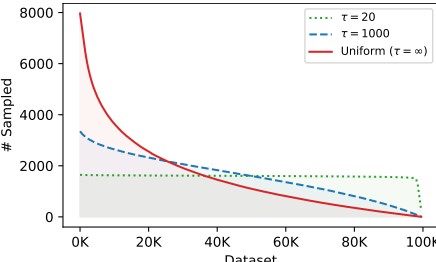 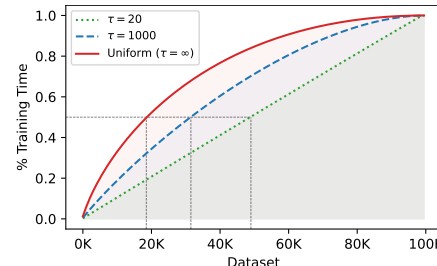

Figure 2: Comparing our balanced dataset sampling procedure (see Equation (6)) for different values of $\tau$ with uniform sampling ($\tau = \infty$). The x-axes correspond to the entries in dataset $\mathcal{D}$ in the order they are experienced. The left plot shows the number of times an entry has been selected for training the world model. The right plot shows the relative amount of training time that has been spent on the data up to that entry. E.g., with uniform sampling, 50% of the training time is used for the first 19K entries, whereas for $\tau = 20$ approximately the same time is spend on both halves of the dataset.

of the policy and it affects the design choices for the world model. Using $x_t = o_t$ (or $\hat{o}_t$) is relatively stable even with imperfect reconstructions $\hat{o}_t$, as the underlying distribution of observations $p(o)$ does not change during training. However, it is also less computationally efficient, since it requires reconstructing the observations during imagination and additional convolutional layers for the policy. Using $x_t = z_t$ (or $\hat{z}_t$) is slightly less stable, as the policy has to adopt to the changes of the distributions $p_\phi(z_t \mid o_t)$ and $p_\psi(\hat{z}_{t+1} \mid h_t)$ during training. Nevertheless, the entropy regularizer and consistency loss in Equation (3) stabilize these distributions. Using $x_t = [z_t, h_t]$ (or $[\hat{z}_t, h_t]$) provides the agent with a summary of the history of experience, but it also adds the burden of running the transformer at inference time. Model-free agents already perform well on most Atari games when using a stack of the most recent frames (e.g., Mnih et al. 2015; Schulman et al. 2017). Therefore, we choose $x_t = z_t$ and apply frame stacking at inference time in order to incorporate short-time information directly into the latent states. At training time we use $x_t = \hat{z}_t$, i.e., the predicted latent states, meaning no frame stacking is applied. As a consequence, our policy is computationally efficient at training time (no reconstructions during imagination) and at inference time (no transformer when running in the real environment).

## 2.3 TRAINING

As is usual for learning with world models, we repeatedly (i) collect experience in the real environment with the current policy, (ii) improve the world model using the past experience, (iii) improve the policy using new experience generated by the world model.

During training we build a dataset $\mathcal{D} = [(o_1, a_1, r_1, d_1), \ldots, (o_T, a_T, r_T, d_T)]$ of the collected experience. After collecting new experience with the current policy, we improve the world model by sampling $N$ sequences of length $\ell$ from $\mathcal{D}$ and optimizing the loss functions in Equations (3) and (4) using stochastic gradient descent. After performing a world model update, we select $M$ of the $N \times \ell$ observations and encode them into latent states to serve as initial states for new trajectories. The dynamics model iteratively generates these $M$ trajectories of length $H$ based on actions provided by the policy. Subsequently, the policy is improved with standard model-free objectives, as described in Section 2.2. In Algorithm 1 we present pseudocode for training the world model and the policy.

**Balanced Dataset Sampling:** Since the dataset grows slowly during training, uniform sampling of trajectories focuses too heavily on early experience, which can lead to overfitting especially in the low data regime. Therefore, we keep visitation counts $v_1, \ldots, v_T$, which are incremented every time an entry is sampled as start of a sequence. These counts are converted to probabilities using the softmax function

$$(p_1, \ldots, p_T) = \mathrm{softmax}\left(-\tfrac{v_1}{\tau}, \ldots, -\tfrac{v_T}{\tau}\right), \tag{6}$$

where $\tau > 0$ is a temperature hyperparameter. With our sampling procedure, new entries in the dataset are oversampled and are selected more often than old ones. Setting $\tau = \infty$ restores uniform sampling as a special case, whereas reducing $\tau$ increases the amount of oversampling. See Figure 2 for a comparison. We empirically show the effectiveness in Section 3.3.

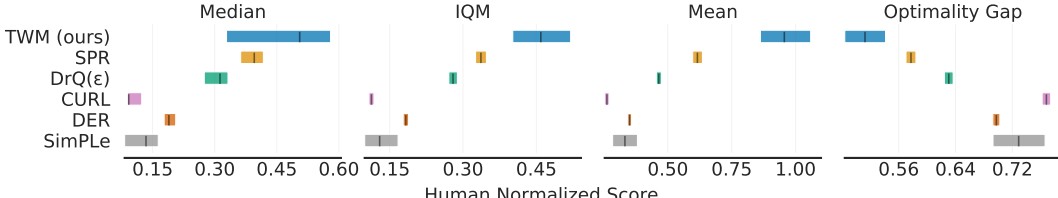

Figure 3: Aggregate metrics on the Atari 100k benchmark with $95\%$ stratified bootstrap confidence intervals (Agarwal et al., 2021). Higher median, interquartile mean (IQM), and mean, but lower optimality gap indicate better performance. Scores for previous methods are from Agarwal et al. (2021) with 100 runs per game (except SimPLe with 5 runs). We evaluate 5 runs per game, leading to wider confidence intervals.

## 3 EXPERIMENTS

To compare data-efficient reinforcement learning algorithms, Kaiser et al. (2020) proposed the Atari 100k benchmark, which uses a subset of 26 Atari games from the Arcade Learning Environment (Bellemare et al., 2013) and limits the number of interactions per game to 100K. This corresponds to 400K frames (because of frame skipping) or roughly 2 hours of gameplay, which is 500 times less than the usual 200 million frames (e.g., Mnih et al. 2015; Schulman et al. 2017; Hafner et al. 2021).

We compare our method with five strong competitors on the Atari 100k benchmark: (i) SimPLe (Kaiser et al., 2020) implements a world model as an action-conditional video prediction model and trains a policy with PPO (Schulman et al., 2017), (ii) DER (van Hasselt et al., 2019) is a variant of Rainbow (Hessel et al., 2018) fine-tuned for sample efficiency, (iii) CURL (Laskin et al., 2020b) improves representations using contrastive learning as an auxiliary task and is combined with DER, (iv) DrQ (Yarats et al., 2021) improves DQN by averaging Q-value estimates over multiple data augmentations of observations, and (v) SPR (Schwarzer et al., 2021) forces representations to be consistent across multiple time steps and data augmentations by extending Rainbow with a self-supervised consistency loss.

### 3.1 RESULTS

We follow the advice of Agarwal et al. (2021) who found significant discrepancies between reported point estimates of mean (and median) scores and a thorough statistical analysis that includes statistical uncertainty. Thus, we report confidence interval estimates of the aggregate metrics median, interquartile mean (IQM), mean, and optimality gap in Figure 3 and performance profiles in Figure 4, which we created using the toolbox provided by Agarwal et al. (2021). The metrics are computed on human normalized scores, which are calculated as `(score_agent – score_random)/` `(score_human – score_random)`. We report the unnormalized scores per game in Table 1. We compare with new scores for DER, CURL, DrQ, and SPR that were evaluated on 100 runs and provided by Agarwal et al. (2021). They report scores for the improved DrQ($\varepsilon$), which is DrQ evaluated with stan-

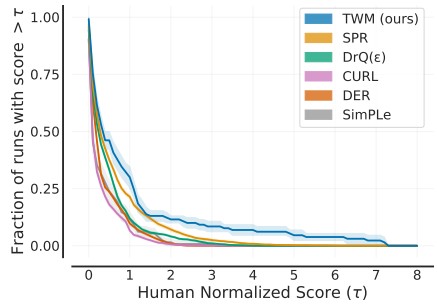

Figure 4: Performance profiles on the Atari 100k benchmark based on score distributions (Agarwal et al., 2021). It shows the fraction of runs across all games (y-axis) above a human normalized score (x-axis). Shaded regions show pointwise $95\%$ confidence bands.

dard $\varepsilon$-greedy parameters. We perform 5 runs per game and compute the average score over 100 episodes at the end of training for each run. TWM shows a significant improvement over previous approaches in all four aggregate metrics and brings the optimality gap closer to zero.

Table 1: Mean scores on the Atari 100k benchmark per game as well as the aggregated human normalized mean and median. We perform 5 runs per game and compute the average over 100 episodes at the end of training for each run. Bold numbers indicate the best scores.

| | | | Model-free | | | | Imagination | |
| Game | Random | Human | DER | CURL | DrQ($\varepsilon$) | SPR | SimPLe | TWM (ours) |
| --- | --- | --- | --- | --- | --- | --- | --- | --- |
| Alien | 227.8 | 7127.7 | 802.3 | 711.0 | **865.2** | 841.9 | 616.9 | 674.6 |
| Amidar | 5.8 | 1719.5 | 125.9 | 113.7 | 137.8 | **179.7** | 74.3 | 121.8 |
| Assault | 222.4 | 742.0 | 561.5 | 500.9 | 579.6 | 565.6 | 527.2 | **682.6** |
| Asterix | 210.0 | 8503.3 | 535.4 | 567.2 | 763.6 | 962.5 | **1128.3** | 1116.6 |
| BankHeist | 14.2 | 753.1 | 185.5 | 65.3 | 232.9 | 345.4 | 34.2 | **466.7** |
| BattleZone | 2360.0 | 37187.5 | 8977.0 | 8997.8 | 10165.3 | **14834.1** | 4031.2 | 5068.0 |
| Boxing | 0.1 | 12.1 | -0.3 | 0.9 | 9.0 | 35.7 | 7.8 | **77.5** |
| Breakout | 1.7 | 30.5 | 9.2 | 2.6 | 19.8 | 19.6 | 16.4 | **20.0** |
| ChopperCommand | 811.0 | 7387.8 | 925.9 | 783.5 | 844.6 | 946.3 | 979.4 | **1697.4** |
| CrazyClimber | 10780.5 | 35829.4 | 34508.6 | 9154.4 | 21539.0 | 36700.5 | 62583.6 | **71820.4** |
| DemonAttack | 152.1 | 1971.0 | 627.6 | 646.5 | **1321.5** | 517.6 | 208.1 | 350.2 |
| Freeway | 0.0 | 29.6 | 20.9 | 20.3 | **28.3** | 19.3 | 16.7 | 24.3 |
| Frostbite | 65.2 | 4334.7 | 871.0 | 1226.5 | 1014.2 | 1170.7 | 236.9 | **1475.6** |
| Gopher | 257.6 | 2412.5 | 467.0 | 400.9 | 621.6 | 660.6 | 596.8 | **1674.8** |
| Hero | 1027.0 | 30826.4 | 6226.0 | 4987.7 | 4167.9 | 5858.6 | 2656.6 | **7254.0** |
| Jamesbond | 29.0 | 302.8 | 275.7 | 331.0 | 349.1 | **366.5** | 100.5 | 362.4 |
| Kangaroo | 52.0 | 3035.0 | 581.7 | 740.2 | 1088.4 | **3617.4** | 51.2 | 1240.0 |
| Krull | 1598.0 | 2665.5 | 3256.9 | 3049.2 | 4402.1 | 3681.6 | 2204.8 | **6349.2** |
| KungFuMaster | 258.5 | 22736.3 | 6580.1 | 8155.6 | 11467.4 | 14783.2 | 14862.5 | **24554.6** |
| MsPacman | 307.3 | 6951.6 | 1187.4 | 1064.0 | 1218.1 | 1318.4 | 1480.0 | **1588.4** |
| Pong | -20.7 | 14.6 | -9.7 | -18.5 | -9.1 | -5.4 | 12.8 | **18.8** |
| PrivateEye | 24.9 | 69571.3 | 72.8 | 81.9 | 3.5 | 86.0 | 35.0 | **86.6** |
| Qbert | 163.9 | 13455.0 | 1773.5 | 727.0 | 1810.7 | 866.3 | 1288.8 | **3330.8** |
| RoadRunner | 11.5 | 7845.0 | 11843.4 | 5006.1 | 11211.4 | **12213.1** | 5640.6 | 9109.0 |
| Seaquest | 68.4 | 42054.7 | 304.6 | 315.2 | 352.3 | 558.1 | 683.3 | **774.4** |
| UpNDown | 533.4 | 11693.2 | 3075.0 | 2646.4 | 4324.5 | 10859.2 | 3350.3 | **15981.7** |
| Normalized Mean | 0.000 | 1.000 | 0.350 | 0.261 | 0.465 | 0.616 | 0.332 | **0.956** |
| Normalized Median | 0.000 | 1.000 | 0.189 | 0.092 | 0.313 | 0.396 | 0.134 | **0.505** |

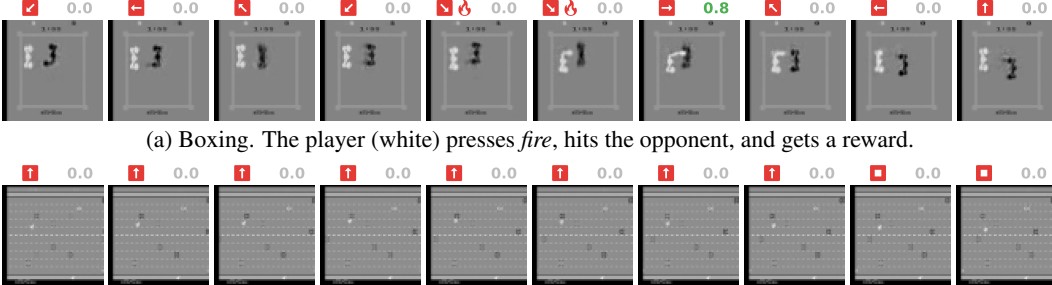

(a) Boxing. The player (white) presses *fire*, hits the opponent, and gets a reward.

(b) Freeway. The player moves up and bumps into a car. The world model correctly pushes the player down, although *up* is still pressed. The movement of the cars is modeled correctly.

Figure 5: Trajectories imagined by our world model. Above each frame we show the performed action and the produced reward.

## 3.2 ANALYSIS

In Figure 5 we show imagined trajectories of our world model. In Figure 6 we visualize an attention map of the transformer for an imagined sequence. In this example a lot of weight is put on the current action and the last three states. However, the transformer also attends to states and rewards in the past, with only past actions being mostly ignored. The two high positive rewards also get high attention, which confirms that the rewards in the input sequence are used by the world model. We hypothesize that these rewards correspond to some events that happened in the environment and this information can be useful for prediction.

An extended analysis can be found in Appendix A.1, including more imagined trajectories and attention maps (and a description of the generation of the plots), sample efficiency, stochasticity of the world model, long sequence imagination, and frame stacking.

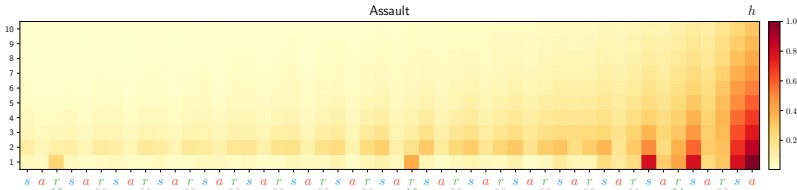

Figure 6: Attention map of the learned transformer for the current hidden state $h$, computed on an imagined trajectory for the game Assault. The x-axis corresponds to the input sequence with the three modalities (states, actions, rewards), where the two rightmost columns are the current state and action. The y-axis corresponds to the layer of the transformer.

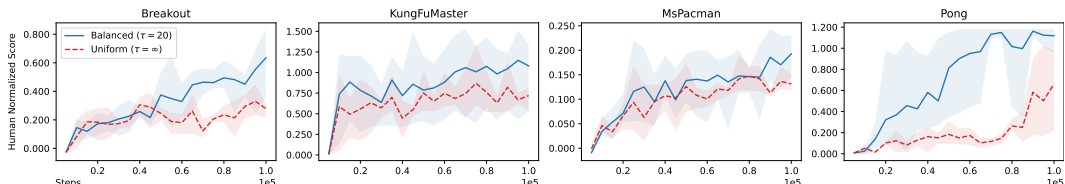

Figure 7: Comparison of the proposed balanced sampling procedure with uniform sampling on a random subset of games. We show the development of the human normalized score in the course of training. The score is higher with balanced sampling, demonstrating its importance.

### 3.3 ABLATION STUDIES

**Uniform Sampling:** To show the effectiveness of the sampling procedure described in Section 2.3, we evaluate three games with uniform dataset sampling, which is equivalent to setting $\tau = \infty$ in Equation (6). In Figure 7 we show that balanced dataset sampling significantly improves the performance in these games. At the end of training, the dynamics loss from Equation (4) is lower when applying balanced sampling. One reason might be that the world model overfits on early training data and performs bad in later stages of training.

**No Rewards:** As described in Section 2.1, the predicted rewards are fed back into the transformer. In Figure 8 we show on three games that this can significantly increase the performance. In some games the performance is equivalent, probably because the world model can make correct predictions solely based on the latent states and actions.

In Appendix A.1 we perform additional ablation studies, including the thresholded entropy loss, a shorter history length, conditioning the policy on $[z, h]$, and increasing the sample efficiency.

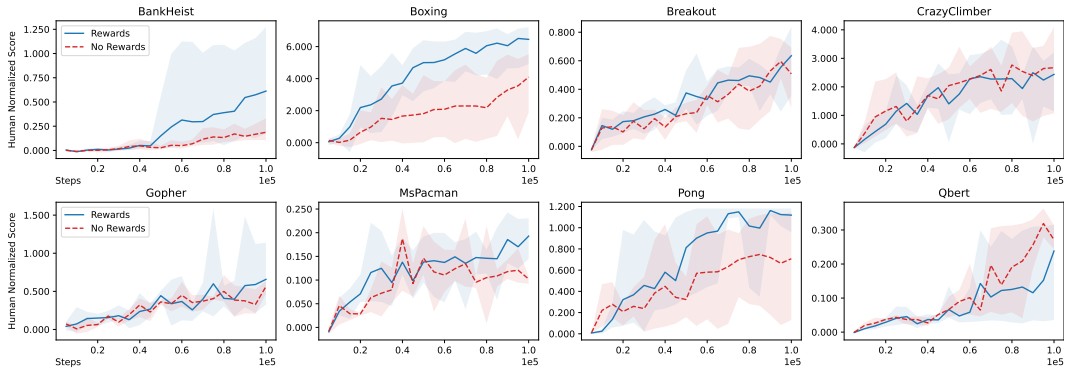

Figure 8: Effect of removing rewards from the input. We show the human normalized score during training of a random subset of games. Conditioning on rewards can significantly increase the performance. Some games do not benefit from the rewards and the score stays roughly the same.

## 4 RELATED WORK

The Dyna architecture (Sutton, 1991) introduced the idea of training a model of the environment and using it to further improve the value function or the policy. Ha & Schmidhuber (2018) introduced the notion of a *world model*, which tries to completely imitate the environment and is used to generate experience to train a model-free agent. They implement a world model as a VAE (Kingma & Welling, 2014) and an RNN and learn a policy in latent space with an evolution strategy. With SimPLe, Kaiser et al. (2020) propose an iterative training procedure that alternates between training the world model and the policy. Their policy operates on pixel-level and is trained using PPO (Schulman et al., 2017). Hafner et al. (2020) present Dreamer and implement a world model as a stochastic RNN that splits the latent state in a stochastic part and a deterministic part; this idea was first introduced by Hafner et al., 2019. This allows their world model to capture the stochasticity of the environment and simultaneously facilitates remembering information over multiple time steps. Robine et al. (2020) use a VQ-VAE to construct a world model with drastically lower number of parameters. DreamerV2 (Hafner et al., 2021) achieves great performance on the Atari 50M benchmark after making some changes to Dreamer, the most important ones being categorical latent variables and an improved objective.

Another direction of model-based reinforcement learning is planning, where the model is used at inference time to improve the action selection by looking ahead several time steps into the future. The most prominent work is MuZero (Schrittwieser et al., 2019), where a learned sequence model of rewards and values is combined with Monte-Carlo Tree Search (Coulom, 2006) without learning explicit representations of the observations. MuZero achieves impressive performance on the Atari 50M benchmark, but it is also computationally expensive and requires significant engineering effort. EfficientZero (Ye et al., 2021) improves MuZero and achieves great performance on the Atari 100k benchmark.

Transformers (Vaswani et al., 2017) advanced the effectiveness of sequence models in multiple domains, such as natural language processing and computer vision (Dosovitskiy et al., 2021). Recently, they have also been applied to reinforcement learning tasks. The Decision Transformer (Chen et al., 2021) and the Trajectory Transformer (Janner et al., 2021) are trained on an offline dataset of trajectories. The Decision Transformer is conditioned on states, actions, and returns, and outputs optimal actions. The Trajectory Transformer trains a sequence model of states, actions, and rewards, and is used for planning. Chen et al. (2022) replace the RNN of Dreamer with a transformer and outperform Dreamer on Hidden Order Discovery tasks. However, their transformer has no access to previous rewards and they do not evaluate their method on the Atari 100k benchmark. Moreover, their policy depends on the outputs of the transformer, leading to higher computational costs during inference time. Concurrent to and independent from our work, Micheli et al. (2022) apply a transformer to sequences of frame tokens and actions and achieve state-of-the-art results on the Atari 100k benchmark.

## 5 CONCLUSION

In this work, we discuss a reinforcement learning approach using transformer-based world models. Our method (TWM) outperforms previous model-free and model-based methods in terms of human normalized score on the 26 games of the Atari 100k benchmark. By using the transformer only during training, we were able to keep the computational costs low during inference, i.e., when running the learned policy in a real environment. We show how feeding back the predicted rewards into the transformer is beneficial for learning the world model. Furthermore, we introduce the balanced cross-entropy loss for finer control over the trade-off between the entropy and cross-entropy terms in the loss functions of the world model. A new thresholded entropy loss effectively stabilizes the entropy of the policy. Finally, our novel balanced sampling procedure corrects issues of naive uniform sampling of past experience.

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

# A APPENDIX

## A.1 EXTENDED EXPERIMENTS

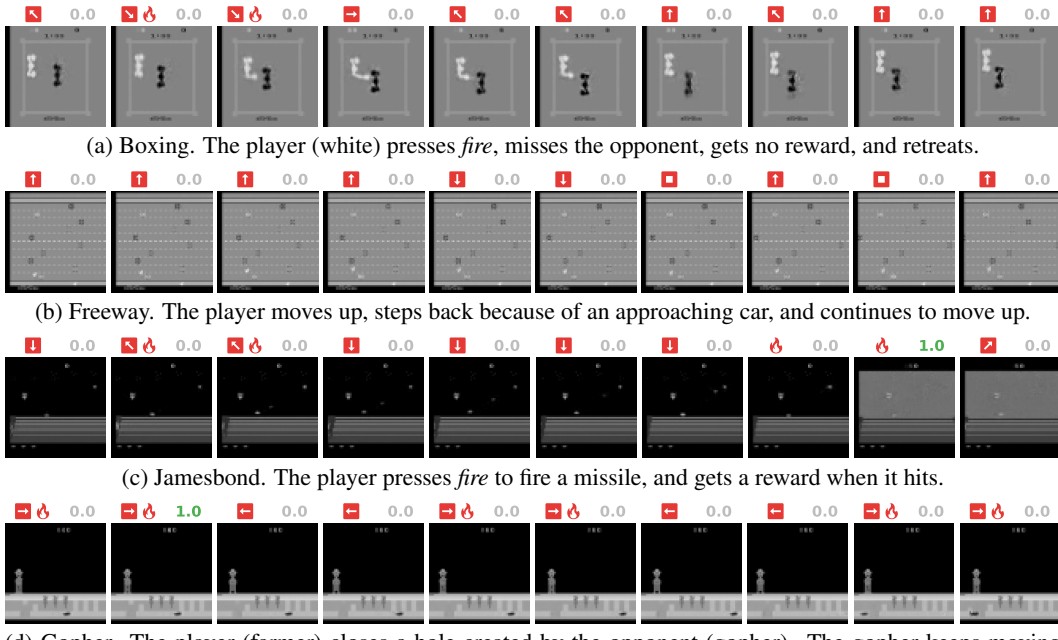

(a) Boxing. The player (white) presses *fire*, misses the opponent, gets no reward, and retreats.

(b) Freeway. The player moves up, steps back because of an approaching car, and continues to move up.

(c) Jamesbond. The player presses *fire* to fire a missile, and gets a reward when it hits.

(d) Gopher. The player (farmer) closes a hole created by the opponent (gopher). The gopher keeps moving independent from the selected actions, indicating that the world model has correctly learned the correlations between the player and the actions. Note that the gopher briefly disappears and reappears on the other side.

Figure 9: Additional example trajectories generated by our world model.

**Additional Analysis:**

1. We provide more example trajectories in Figure 9.

2. We present more attention plots in Figures 10 and 11. All attention maps are generated using the attention rollout method by Abnar & Zuidema (2020). Note that we had to modify the method slightly, in order to take the causal masks into account.

3. *Sample Efficiency*: We provide the scores of our main experiments after different amounts of interactions with the environment in Table 2. After 50K interactions, our method already has a higher mean normalized score than previous sample-efficient methods. Our mean normalized score is higher than DER, CURL, and SimPLe after 25K interactions. This demonstrates the high sample efficiency of our approach.

4. *Stochasticity*: The stochastic prediction of the next state allows the world model to sample a variety of trajectories, even from the same starting state, as can be seen in Figure 12.

5. *Long Sequence Imagination*: The world model is trained using sequences of length $\ell = 16$, however, it generalizes well to very long trajectories, as shown in Figure 13.

6. *Frame Stacking*: In Figure 14 we visualize the learned stacks of frames. This shows that the world model encodes and predicts the motion of objects.

**Additional Ablation Studies:**

1. *Thresholded Entropy Loss*: In Figure 15 we compare (i) our thresholded entropy loss for the policy (see Section 2.2) with (ii) the usual entropy penalty. For (i) we use the same hyperparameters as in our main experiments, i.e., $\eta = 0.01$ and $\Gamma = 0.1$. For (ii) we set $\eta = 0.001$ and $\Gamma = 1.0$, which effectively disables the threshold. Without a threshold, the entropy is more likely to either collapse or diverge. When the threshold is used, the score is

higher as well, probably because the entropy is in a more sensible range for the exploration-exploitation trade-off. This cannot be solved by adjusting the penalty coefficient $\eta$ alone, since it would increase or decrease the entropy in all games.

2. *History Length*: We trained our world model with a shorter history and set $\ell = 4$ instead of $\ell = 16$. This has a negative impact on the score, as can be seen in Figure 16, demonstrating that more time steps into the past are important.

3. *Choice of Policy Input*: In Section 2.2 we explained why the input to the policy is only the latent state, i.e., $x = z$. In Figure 17 we show that using $x = [z, h]$ can result in lower final scores. We hypothesize that the policy network has a hard time keeping up with the changes of the space of $h$ during training and cannot ignore this additional information.

4. *Increasing the Sample Efficiency*: To find out whether we can further increase the sample efficiency shown in Table 2, we train a random subset of games again on 10K, 25K, and 50K interactions with the full training budget that we used for the 100K interactions. In Figure 18 we see that this can lead to significant improvements in some cases, which could mean that the policy benefits from more training on imagined trajectories, but can even lead to worse performance in other cases, which could possibly be caused by overfitting of the world model. When the performance stays the same even with longer training, this could mean that better exploration in the real environment is required to get further improvements.

Table 2: Performance of our method at different stages of training compared with final scores of previous methods. We show individual game scores and mean human normalized scores. The normalized mean of our method is higher than SimPLe after only 25K interactions, and higher than previous methods after 50K interactions.

| Game | Random | Human | SimPLe | SPR | TWM (ours) | | | | | |
|---|---|---|---|---|---|---|---|---|---|---|
| | | | | | 5K | 10K | 25K | 50K | 75K | 100K |
| Alien | 227.8 | 7127.7 | 616.9 | 841.9 | 202.8 | 383.2 | 463.6 | 532.0 | 776.6 | 674.6 |
| Amidar | 5.8 | 1719.5 | 74.3 | 179.7 | 3.8 | 35.4 | 54.9 | 101.3 | 103.0 | 121.8 |
| Assault | 222.4 | 742.0 | 527.2 | 565.6 | 241.5 | 315.4 | 418.7 | 466.8 | 627.8 | 682.6 |
| Asterix | 210.0 | 8503.3 | 1128.3 | 962.5 | 277.0 | 297.0 | 536.0 | 912.0 | 886.0 | 1116.6 |
| BankHeist | 14.2 | 753.1 | 34.2 | 345.4 | 17.6 | 4.4 | 17.4 | 125.2 | 288.4 | 466.7 |
| BattleZone | 2360.0 | 37187.5 | 4031.2 | 14834.1 | 2640.0 | 3120.0 | 2700.0 | 3740.0 | 5260.0 | 5068.0 |
| Boxing | 0.1 | 12.1 | 7.8 | 35.7 | 0.8 | 3.4 | 28.5 | 60.1 | 67.1 | 77.5 |
| Breakout | 1.7 | 30.5 | 16.4 | 19.6 | 1.0 | 5.9 | 6.9 | 12.5 | 15.0 | 20.0 |
| ChopperCommand | 811.0 | 7387.8 | 979.4 | 946.3 | 928.0 | 1044.0 | 1358.0 | 1306.0 | 1438.0 | 1697.4 |
| CrazyClimber | 10780.5 | 35829.4 | 62583.6 | 36700.5 | 7425.0 | 14773.2 | 39456.8 | 45916.0 | 67766.2 | 71820.4 |
| DemonAttack | 152.1 | 1971.0 | 208.1 | 517.6 | 174.7 | 184.4 | 216.8 | 335.2 | 391.4 | 350.2 |
| Freeway | 0.0 | 29.6 | 16.7 | 19.3 | 0.0 | 4.6 | 20.8 | 23.7 | 23.9 | 24.3 |
| Frostbite | 65.2 | 4334.7 | 236.9 | 1170.7 | 66.2 | 204.6 | 297.8 | 247.6 | 1165.4 | 1475.6 |
| Gopher | 257.6 | 2412.5 | 596.8 | 660.6 | 345.2 | 414.0 | 593.2 | 1213.2 | 1549.2 | 1674.8 |
| Hero | 1027.0 | 30826.4 | 2656.6 | 5858.6 | 448.9 | 1552.6 | 4790.9 | 6302.7 | 9403.8 | 7254.0 |
| Jamesbond | 29.0 | 302.8 | 100.5 | 366.5 | 35.0 | 117.0 | 172.0 | 215.0 | 322.0 | 362.4 |
| Kangaroo | 52.0 | 3035.0 | 51.2 | 3617.4 | 28.0 | 92.0 | 476.0 | 724.0 | 876.0 | 1240.0 |
| Krull | 1598.0 | 2665.5 | 2204.8 | 3681.6 | 1763.6 | 2552.8 | 4234.0 | 4699.2 | 5848.0 | 6349.2 |
| KungFuMaster | 258.5 | 22736.3 | 14862.5 | 14783.2 | 574.0 | 16828.0 | 16368.0 | 17946.0 | 22936.0 | 24554.6 |
| MsPacman | 307.3 | 6951.6 | 1480.0 | 1318.4 | 245.9 | 535.1 | 1077.5 | 1224.3 | 1287.6 | 1588.4 |
| Pong | -20.7 | 14.6 | 12.8 | -5.4 | -20.4 | -19.8 | -7.7 | 8.0 | 19.9 | 18.8 |
| PrivateEye | 24.9 | 69571.3 | 35.0 | 86.0 | 61.0 | 80.0 | 80.0 | 3.2 | 88.8 | 86.6 |
| Qbert | 163.9 | 13455.0 | 1288.8 | 866.3 | 151.0 | 298.5 | 703.5 | 1046.5 | 1788.5 | 3330.8 |
| RoadRunner | 11.5 | 7845.0 | 5640.6 | 12213.1 | 24.0 | 1120.0 | 5178.0 | 7436.0 | 8034.0 | 9109.0 |
| Seaquest | 68.4 | 42054.7 | 683.3 | 558.1 | 76.8 | 221.2 | 428.4 | 572.0 | 704.0 | 774.4 |
| UpNDown | 533.4 | 11693.2 | 3350.3 | 10859.2 | 385.8 | 1963.0 | 2905.6 | 4922.8 | 10478.6 | 15981.7 |
| Normalized Mean | 0.000 | 1.000 | 0.332 | 0.616 | 0.007 | 0.133 | 0.408 | 0.624 | 0.832 | 0.956 |

**Wall-Clock Times:** For each run, we give the agent a total training and evaluation budget of roughly 10 hours on a single NVIDIA A100 GPU. The time can vary slightly, since the budget is based on the number of updates. An NVIDIA GeForce RTX 3090 requires 12-13 hours for the same amount of training and evaluation. When using a vanilla transformer, which does not use the memory mechanism of the Transformer-XL architecture (Dai et al., 2019), the runtime is roughly 15.5 hours on an NVIDIA A100 GPU, i.e., 1.5 times higher.

We compare the runtime of our method with previous methods in Table 3. Our method is more than 20 times faster than SimPLe, but slower than model-free methods. However, our method should be as fast as other model-free methods during inference. In Table 2 we have shown that our method

Table 3: Approximate runtime (i.e., training and evaluation time for a single run) of our method compared with previous methods that also evaluate on the Atari 100k benchmark. Runtimes of previous methods are taken from Schwarzer et al. (2021). They used an improved version of DER (van Hasselt et al., 2019), which is roughly equivalent to DrQ (Yarats et al., 2021), so the specified runtime might differ from the original DER implementation. There are data augmented versions for SPR and DER. All runtimes are measured on a single NVIDIA P100 GPU.

| Method | Model-based | Runtime in hours |
|---|---|---|
| SimPLe | ✓ | 500 |
| TWM (ours) | ✓ | 23.3 |
| SPR (with aug.) | ✗ | 4.6 |
| SPR (w/o aug.) | ✗ | 3.0 |
| DER/DrQ (with aug.) | ✗ | 2.1 |
| DER/DrQ (w/o aug.) | ✗ | 1.4 |

achieves a higher human normalized score than previous sample-efficient methods after 50K interactions. This suggests that our method could potentially outperform previous methods with shorter training, which would take less than 23.3 hours.

To determine how time-consuming the individual parts of our method are, we investigate the throughput of the models, with the batch sizes of our main experiments. The Transformer-XL version is almost twice as fast, which again shows the importance of this design choice. The throughputs were measured on an NVIDIA A100 GPU and are given in (approximate) samples per second:

- World model training: 16,800 samples/s
- World model imagination (Transformer-XL): 39,000 samples/s
- World model imagination (vanilla): 19,900 samples/s
- Policy training: 700,000 samples/s

We also examine how fast the policy can run in an Atari game. We measured the (approximate) frames per second on a CPU (since the batch size is 1). Conditioning the policy on $[z, h]$ is about 3 times slower than $z$, since the transformer is required:

- Policy conditioned on $z$: 653 frames/s
- Policy conditioned on $[z, h]$: 213 frames/s

## A.2 DERIVATION OF BALANCED CROSS-ENTROPY LOSS

Hafner et al. (2021) propose to use a balanced KL divergence loss to jointly optimize the observation encoder $q_\theta$ and state predictor $p_\theta$ with shared parameters $\theta$, i.e.,

$$\lambda\, D_{\mathrm{KL}}(\mathrm{sg}(q_\theta) \parallel p_\theta) + (1 - \lambda)\, D_{\mathrm{KL}}(q_\theta \parallel \mathrm{sg}(p_\theta)), \tag{7}$$

where $\mathrm{sg}(\cdot)$ denotes the stop-gradient operation and $\lambda \in [0, 1]$ controls how much the state predictor adapts to the observation encoder and vice versa. We use the identity $D_{\mathrm{KL}}(q \parallel p) = H(q, p) - H(q)$, where $H(q, p)$ is the cross-entropy of distribution $p$ relative to distribution $q$, and show that our loss functions lead to the same gradients as the balanced KL objective, but with finer control over the individual components:

$$\nabla_\theta \left[ \lambda\, D_{\mathrm{KL}}(\mathrm{sg}(q_\theta) \parallel p_\theta) + (1 - \lambda)\, D_{\mathrm{KL}}(q_\theta \parallel \mathrm{sg}(p_\theta)) \right] \tag{8}$$

$$= \nabla_\theta \left[ \lambda\, (H(\mathrm{sg}(q_\theta), p_\theta) - H(\mathrm{sg}(q_\theta))) + (1 - \lambda)\, (H(q_\theta, \mathrm{sg}(p_\theta)) - H(q_\theta)) \right] \tag{9}$$

$$= \nabla_\theta \left[ \lambda_1\, H(\mathrm{sg}(q_\theta), p_\theta) + \lambda_2\, H(q_\theta, \mathrm{sg}(p_\theta)) - \lambda_3\, H(q_\theta) \right], \tag{10}$$

since $\nabla_\theta H(\mathrm{sg}(q_\theta)) = 0$ and by defining $\lambda_1 = \lambda$ and $\lambda_2 = 1 - \lambda$ and $\lambda_3 = 1 - \lambda$. In this form, we have control over the cross-entropy of the state predictor relative to the observation encoder and vice versa. Moreover, we explicitly penalize the entropy of the observation encoder, instead of being entangled inside of the KL divergence.

As common in the literature, we define the loss function by omitting the gradient in Equation (10), so that automatic differentiation computes this gradient. For our world model, we split the objective into two loss functions, as the observation encoder and state predictor have separate parameters, yielding Equations (3) and (4).

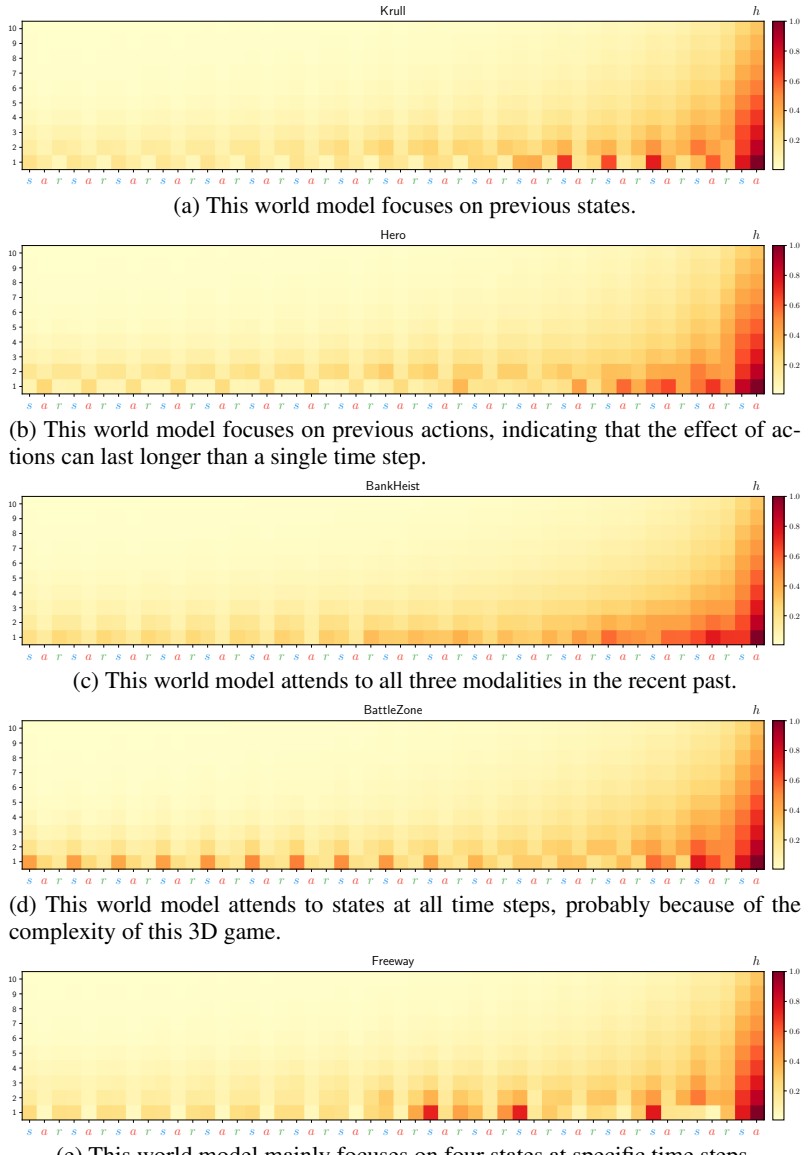

(a) This world model focuses on previous states.

(b) This world model focuses on previous actions, indicating that the effect of actions can last longer than a single time step.

(c) This world model attends to all three modalities in the recent past.

(d) This world model attends to states at all time steps, probably because of the complexity of this 3D game.

(e) This world model mainly focuses on four states at specific time steps.

Figure 10: Average attention maps of the transformer, computed over many time steps. They show how different games require a different focus on modalities and time steps.

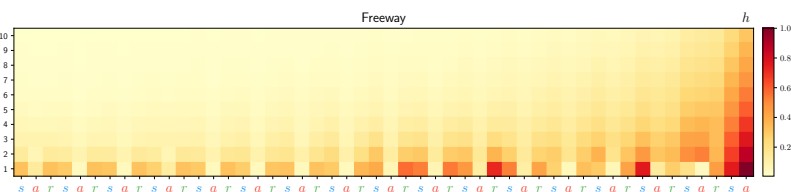

Figure 11: Attention map for Freeway for a single time step. At this point the player hits a car and gets pushed back (see also Figure 5b) and the world model puts more attention to past states and rewards, compared with the average attention at other time steps, as shown in Figure 10e. The world model has learned to handle this situation separately.

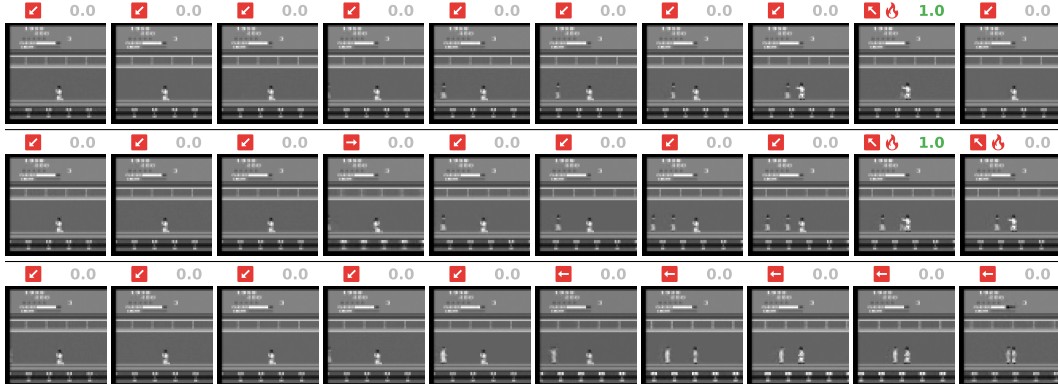

Figure 12: Three trajectories for the game KungFuMaster generated by our world model, using the same starting state. Because of its stochastic nature, the world model is able to generate three different situations (one opponent, two opponents, one other type of opponent). Note that we only show every third frame to cover more time steps.

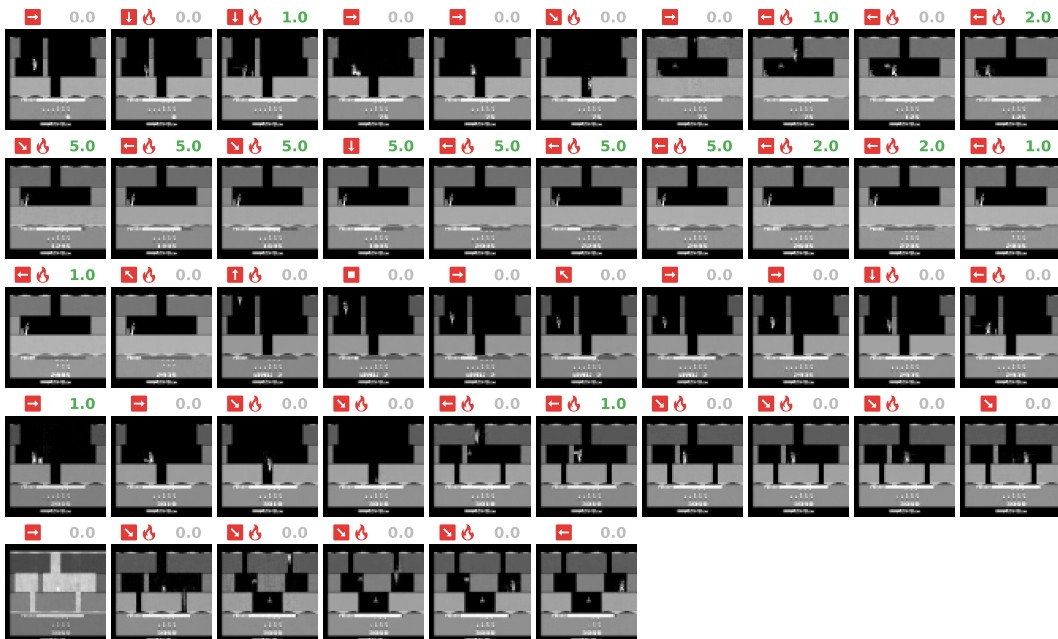

Figure 13: A long trajectory imagined by our world model for the game Hero. The player traverses five different rooms and the world model is able to correctly predict the state and reward dynamics. Note that we only show every fifth frame to cover more time steps (the rewards lying in-between are summed up). The total number of time steps is 230.

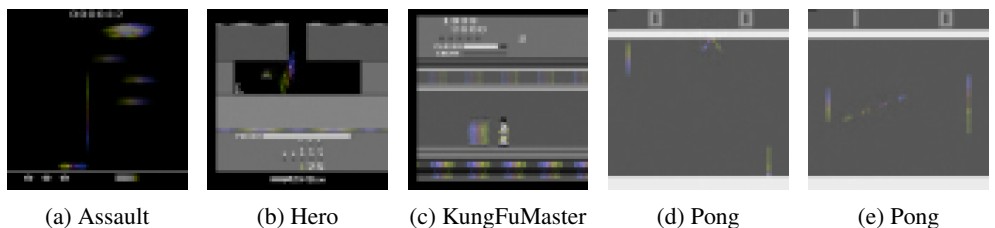

| (a) Assault | (b) Hero | (c) KungFuMaster | (d) Pong | (e) Pong |

Figure 14: Visualization of frame stacks reconstructed from predicted states $\hat{z}_t$. Each frame in the stack is visualized by a different color. The world model is able to encode and predict movements.

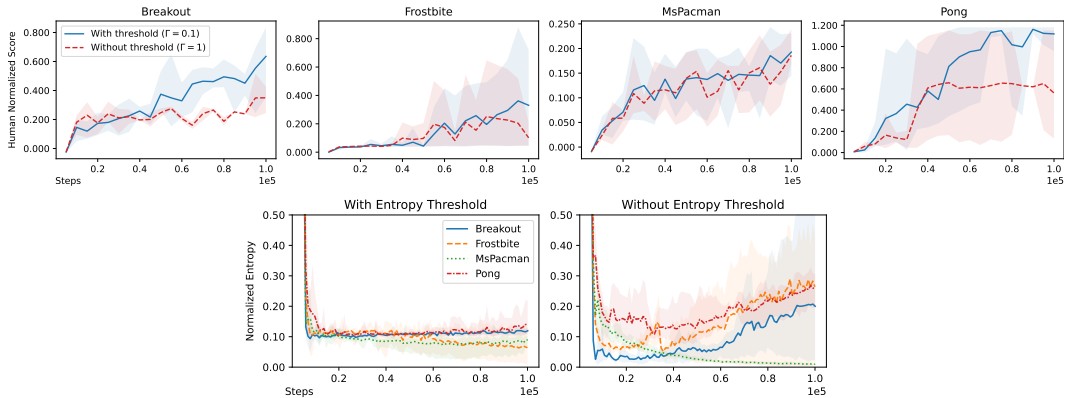

Figure 15: Effect of disabling the proposed thresholded entropy loss (by setting $\Gamma = 1$) on the performance and the entropy in a random subset of games. The thresholded version stabilizes the entropy and leads to a better score in Breakout and Pong, while the entropy behaves unfavorably without a threshold.

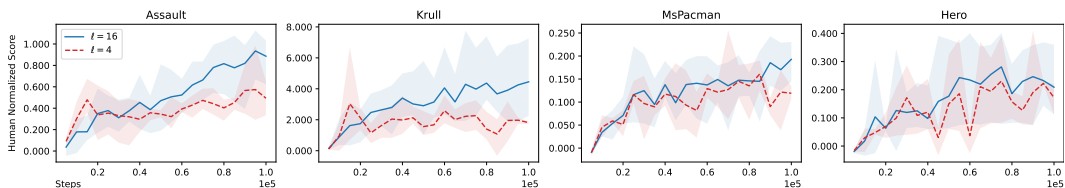

Figure 16: Comparison of the history length $\ell = 16$ used in our main experiments with $\ell = 4$ on a random subset of games. We observe a lower human normalized score for $\ell = 4$.

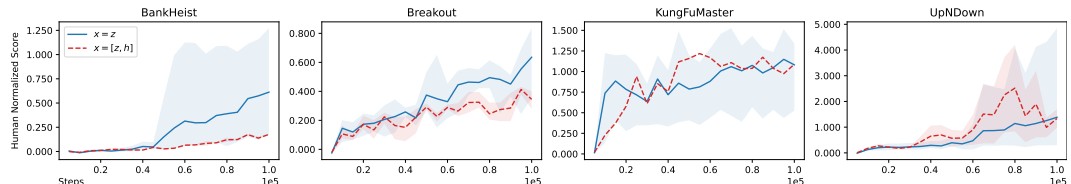

Figure 17: Conditioning the policy on $[z, h]$ compared with the usual $z$. In some cases the performance can be better during training, but the final score is lower or equal.

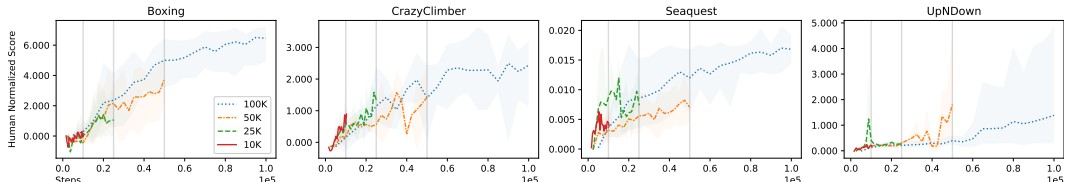

Figure 18: Scores on a random subset of games when we train with a lower number of interactions but the same training budget. This only leads to a significant improvement for UpNDown, where the final score is higher with only 50K interactions.

Figure 19: Comparison of the proposed balanced sampling procedure with uniform sampling. It shows the development of the dynamics loss from Equation (4), which is lower at the end of training in all cases.

## A.3 ADDITIONAL TRAINING DETAILS

In Algorithm 1 we present pseudocode for training the world model and the actor-critic agent. We use the SiLU activation function (Elfwing et al., 2018) for all models. In Table 4 we summarize all hyperparameters that we used in our experiments. In Table 5 we provide the number of parameters of our models.

**Pretraining for Better Initialization:** During training we need to correctly balance the amount of world model training and policy training, since the policy has to keep up with the distributional shift of the latent space. However, we can spend some extra training time on the world model with pre-collected data (included in the 100K interactions) at the beginning of training in order to obtain a reasonable initialization for the latent states.

---

**Algorithm 1** Training the world model and the actor-critic agent.

```
function train_world_model()             function train_actor_critic(z)
   // sample sequences of observations,     // imagine trajectories of states,
   // rewards, actions and discounts        // rewards, actions and discounts;
   o,a,r,d = sample_from_dataset()          // use z as starting point
   z = encode(o)                            imag = [z]
   o_hat = decode(z)                        for t = 0 until H do
   h = transformer(z,a,r)                      a = actor(z)
   r_hat,d_hat,z_hat = predict(h)              imag.append(a)
                                               h = transformer(imag)
   // optimize world model via                 r,d,z = predict(h)
   // self-supervised learning                 imag.extend([r,d,z])
   optim_observation(o,z,o_hat,z_hat)
   optim_dynamics(r,d,z,r_hat,d_hat,z_hat)  // optimize actor-critic via
                                            // reinforcement learning
   // z will be used for imagination        optim_actor_critic(imag)
   return z
```

---

Table 4: Hyperparameters used in our experiments.

| Description | Symbol | Value |
|---|---|---|
| Dataset sampling temperature | $\tau$ | 20 |
| Discount factor | $\gamma$ | 0.99 |
| GAE parameter | $\lambda$ | 0.95 |
| World model batch size | $N$ | 100 |
| History length | $\ell$ | 16 |
| Imagination batch size | $M$ | 400 |
| Imagination horizon | $H$ | 15 |
| Encoder entropy coefficient | $\alpha_1$ | 5.0 |
| Consistency loss coefficient | $\alpha_2$ | 0.01 |
| Reward coefficient | $\beta_1$ | 10.0 |
| Discount coefficient | $\beta_2$ | 50.0 |
| Actor entropy coefficient | $\eta$ | 0.01 |
| Actor entropy threshold | $\Gamma$ | 0.1 |
| Environment steps | — | 100K |
| Frame skip | — | 4 |
| Frame down-sampling | — | $64 \times 64$ |
| Frame gray-scaling | — | Yes |
| Frame stack | — | 4 |
| Terminate on live loss | — | Yes |
| Max frames per episode | — | 108K |
| Max no-ops | — | 30 |
| Observation learning rate | — | 0.0001 |
| Dynamics learning rate | — | 0.0001 |
| Actor learning rate | — | 0.0001 |
| Critic learning rate | — | 0.00001 |
| Transformer embedding size | — | 256 |
| Transformer layers | — | 10 |
| Transformer heads | — | $4 \times 64$ |
| Transformer feedforward size | — | 1024 |
| Latent state predictor units | — | $4 \times 512$ |
| Reward predictor units | — | $4 \times 256$ |
| Discount predictor units | — | $4 \times 256$ |
| Actor units | — | $4 \times 512$ |
| Critic units | — | $4 \times 512$ |
| Activation function | — | SiLU |

Table 5: Number of parameters of our models.

| Model | Symbol | # Parameters |
|---|---|---|
| Observation model | $\phi$ | 8.2M |
| Dynamics model | $\psi$ | 10.8M |
| Actor | $\theta$ | 1.3M |
| Critic | $\xi$ | 1.3M |
| World model | — | 19M |
| Actor-critic | — | 2.6M |
| Total | — | 21.6M |
| Encoder + actor (at inference time) | — | 4.4M |

