# OpenReview forum: "Transformer-based World Models Are Happy With 100k Interactions"
_ICLR.cc/2023/Conference — ICLR 2023 poster_

### Official Review · Reviewer_swpJ · 2022-10-13

**Confidence:** 4
**Correctness:** 4
**Technical Novelty And Significance:** 3
**Empirical Novelty And Significance:** 3
**Recommendation:** 8

**Clarity, Quality, Novelty And Reproducibility:**

It is well-written and clear, and the proposed techniques are densely analyzed through their ablation studies.

In the aspect of the novelty of their modeling, it is not too new, but they first analyzed the Transformer-based world model for the Atari 100k benchmark and showed better performance than the state-of-the-art baselines. Their techniques, balanced dataset sampling, and thresholded entropy loss are pretty new, and they showed those are important to increase performance.

It looks easy to be reproduced.

**Strength And Weaknesses:**

Strength
- The paper is well-written and easy to follow.
- They also support ablation studies for suggested techniques, balanced dataset sampling, threshold entropy loss, and feeding back the predicted reward. Thus, the reader can understand why they applied those theoretically and empirically.
- Using $z_t$ as the input of the policy, not the $h_t$. This choice can interact with real environment more quickly, which is different from previous works [1, 3].

Weakness
- The motivation why the Transformer-based World Model will work well is not considered (e.g., in [1,2,3,4], they address the long-term dependency issue). Maybe better transition modeling could be one of the motivations. Then it should make this paper more concrete if there were empirical results about the world model qualities when using Transformer and recurrent modules.
- As pointed out above, their modeling cleverly uses $z_t$ as an input of policy, while it is limited for attending the history. It could be optional, like for the task requiring long-term dependency, using $h_t$ and usually using $z_t$, but when using $h_t$, the inference could be slower.



**Summary Of The Paper:**

This paper investigates the Transformer-based world model for model-based reinforcement learning.

Inspired by the successful applications of Transformers to supervised learning or generative learning, applying Transformers to reinforcement learning has been one of the most interesting topics among RL community members [1, 2, 3, 4, 5]. However, one of the pain points of Transformer applications is the much larger computation overheads when interacting with a real environment or learning in imagination [1, 2, 3]. The world model design in this paper reduces the overhead while it limits the agent to attend to the past at inference time.

In addition, they suggest reasonable techniques to train the agent on top of this world model to be more stable and better and show the ablation studies with/without the techniques too. Their model outperforms baselines, including the state-of-the-art model-free and model-based methods.



**Summary Of The Review:**

This paper does not consider some points about why Transformer is good for the world model and how their model can handle the long-term dependent knowledge. However, they showed impressive performance by applying Transformer to world model with their techniques, which is good enough to be shared in our community. Especially by densely analyzing their techniques, they made the reader easily understand why the techniques are required. Hopefully, the authors will revise this paper by addressing my concerns.


[1] Parisotto, Emilio, et al. "Stabilizing transformers for reinforcement learning." International conference on machine learning. PMLR, 2020.

[2] Parisotto, Emilio, and Ruslan Salakhutdinov. "Efficient transformers in reinforcement learning using actor-learner distillation." arXiv preprint arXiv:2104.01655 (2021).

[3] Chen, Chang, et al. "Transdreamer: Reinforcement learning with transformer world models." arXiv preprint arXiv:2202.09481 (2022).

[4] Esslinger, Kevin, Robert Platt, and Christopher Amato. "Deep Transformer Q-Networks for Partially Observable Reinforcement Learning." arXiv preprint arXiv:2206.01078 (2022).

[5] Micheli, Vincent, Eloi Alonso, and François Fleuret. "Transformers are sample efficient world models." arXiv preprint arXiv:2209.00588 (2022).

---

> ### Author Response · Authors · 2022-11-17
> **Response to Reviewer swpJ**
>
> We would like to thank the reviewer for reviewing our submission and providing detailed comments.
>
> > The motivation why the Transformer-based World Model will work well is not considered (e.g., in [1,2,3,4], they address the long-term dependency issue). Maybe better transition modeling could be one of the motivations. Then it should make this paper more concrete if there were empirical results about the world model qualities when using Transformer and recurrent modules.
>
> Our motivations for using a transformer are indeed better transition modeling, i.e., it can model more complex dependencies because of the direct access to previous time steps, and the easy integration of previous rewards (as described in Section 2.1, “Autoregressive Dynamics Model”). To better understand how the transformer works, we visualized the learned attention (see Section 3.2 and Appendix A.1).
>
> > As pointed out above, their modeling cleverly uses $z_t$ as an input of policy, while it is limited for attending the history. It could be optional, like for the task requiring long-term dependency, using $h_t$ and usually using $z_t$, but when using $h_t$, the inference could be slower.
>
> You are right that the inference is slower when we include $h_t$ (see new Appendix A.1, “Wall-Clock Times”). However, it will be interesting to investigate this on other tasks in the future.
>
> > This paper does not consider some points about why Transformer is good for the world model and how their model can handle the long-term dependent knowledge.
>
> We added analyses to better understand what the world model has learned and performed a new ablation study regarding the length of the history. We believe that this sheds light onto the question why transformers are particularly good.

---

### Official Review · Reviewer_6X3a · 2022-10-22

**Confidence:** 4
**Correctness:** 2
**Technical Novelty And Significance:** 2
**Empirical Novelty And Significance:** 2
**Recommendation:** 6

**Clarity, Quality, Novelty And Reproducibility:**

### Clarity and Quality

- The paper is clearly written and reads well

### Novelty

While the proposed architecture seems novel but it's not surprisingly novel as there has been an effort to incorporate Transformers into world models [Chen et al, 2022] and there has been recent works that also takes similar two-stage approaches in the context of video prediction [Rakhimov et al., 2020; Yan et al., 2021]. But I acknowledge the details can matter for world models so it could be a novelty of the approach if this can be thoroughly supported with analysis and additional results.

### Reproducibility

Source code is included so it could be reproduced. But more analysis on wall time could be nice in this front.

[Chen et al., 2022] Chen, Chang, Yi-Fu Wu, Jaesik Yoon, and Sungjin Ahn. "Transdreamer: Reinforcement learning with transformer world models." arXiv preprint arXiv:2202.09481 (2022).

[Rakhimov et al., 2020] Rakhimov, Ruslan, Denis Volkhonskiy, Alexey Artemov, Denis Zorin, and Evgeny Burnaev. "Latent video transformer." arXiv preprint arXiv:2006.10704 (2020).

[Yan et al., 2021] Yan, Wilson, Yunzhi Zhang, Pieter Abbeel, and Aravind Srinivas. "Videogpt: Video generation using vq-vae and transformers." arXiv preprint arXiv:2104.10157 (2021).

**Strength And Weaknesses:**

## Strengths

- Strong performance with simple and intuitive method
- Paper reads well

## Weaknesses

- Claims within the paper are not justified enough except that the proposed method performs well.
    - It is not clear whether transformer-based model is indeed better than previous architecture with GRU of DreamerV2. From the current results, we cannot know whether the gain comes from additional techniques or transformer-based world models or additional computes. How does it compare to DreamerV2 trained with the proposed techniques, similar computes (not parameters), and hyperparameters to be more sample-efficient?
    - The paper says training is more efficient as it's parallel, but does not provide any measurement or investigation to explicitly show this compared to previous architecture based on a recurrent architecture. Gviven that the method utilizes a quite large model consisting of 10-layers Transformers, this should be further thoroughly supported.
    - It is written that leveraging Transformer-XL [Dai et al., 2019] can be good for long-term inference, but in the paper it's not investigated whether this is indeed the case. Additional results that compared with vanilla Transformer architecture could be included to support this.
    - It is claimed that long-term dependencies can be captured with Transformers but final policy is trained on top of the representation from observation consisting of 4 stcaked frames. So it's difficult to know it's really good for capturing long-term dependencies in terms of either prediction for generating training samples or inference. It should be evaluated on more challenging tasks that require memories?
    - For thresholded entropy loss, I can understand that it can make normalized entropy stable, but is this necessarily better than changing entropy throughout the training? How does the performance is related to the change in the entropy here?
    - For sampling scheme, it would be nice to further describe and formulate how this cheme would affect the optimization objectives compared to using uniform sampling scheme. Moreover, ablation to see how this affects performance would be also nice.
- Investigation into the effect of conditioning the policy on $z_t$ is not complete. It's okay to say that this would incur more costs, but it seems an important difference to prior approaches, so there should be an investigation into this with additional results and also corresponding wall times.
- Comparison with prior works can be much more informative if it includes some metrics in terms of training costs -- as it seems like the proposed method would require much more resources than other approaches (of course it's my guess because related information is not included anywhere, correct if i'm wrong), and it also does some pre-training on initial exploration samples.
- Missing comparison with TransDreamer, and not clear why this is only included in Related Work?

[Dai et al., 2019] Dai, Zihang, Zhilin Yang, Yiming Yang, Jaime Carbonell, Quoc V. Le, and Ruslan Salakhutdinov. "Transformer-xl: Attentive language models beyond a fixed-length context." arXiv preprint arXiv:1901.02860 (2019).

[Chen et al., 2022] Chen, Chang, Yi-Fu Wu, Jaesik Yoon, and Sungjin Ahn. "Transdreamer: Reinforcement learning with transformer world models." arXiv preprint arXiv:2202.09481 (2022).

---

Updating the score to 6 after the discussion with AC and other reviewers. I would like to strongly encourage the authors to include results to compare with RNN-based world models or tone-down on claims on the benefit of transformer over RNN-based world models.


**Summary Of The Paper:**

The paper introduces a model-based RL approach that learns a transformer-based world model and additional techniques to further improve the sample-efficiency. Specifically, VAE is trained to learn visual information from raw pixels, and autoregressive dynamics model based on Transformer-XL architecture is trained upon a sequence consisting of states and rewards. The method is evaluated on a set of Atari tasks within 100k steps to measure sample-efficiency of the proposed method.

**Summary Of The Review:**

While the paper proposed an intuitive approach that achieves strong sample-efficiency, the main weakness of the paper is in that it lacks analysis and experimental results to support the claims made in the paper, which makes me difficult to see that the current draft is strong enough to be accepted as a ICLR paper (I'm trying to avoid borderline score following the suggestion from the conference this year). Hence I'm recommending the paper to be rejected at the current status but willing to update the score per the discussion within the rebuttal period.

---

> ### Author Response · Authors · 2022-11-17
> **Response to Reviewer 6X3a (continued)**
>
> > Missing comparison with TransDreamer, and not clear why this is only included in Related Work?
>
> Unfortunately, there are no results available for TransDreamer on the Atari 100k benchmark, so we can not include it in the comparison. We already used up our computing budget by evaluating our method.
>
> > While the paper proposed an intuitive approach that achieves strong sample-efficiency, the main weakness of the paper is in that it lacks analysis and experimental results to support the claims made in the paper, which makes me difficult to see that the current draft is strong enough to be accepted as a ICLR paper (I'm trying to avoid borderline score following the suggestion from the conference this year).
>
> We hope that our revised paper (including an analysis of our world model, additional ablation experiments, and wall-clock times) and our responses convince you to recommend acceptance.
>
> **References:**
> [1] Schwarzer, Max, et al. "Data-Efficient Reinforcement Learning with Self-Predictive Representations." International Conference on Learning Representations. 2020.
> [2] Hessel, Matteo, et al. "Rainbow: Combining improvements in deep reinforcement learning." Thirty-second AAAI conference on artificial intelligence. 2018.
> [3] Kaiser, Łukasz, et al. "Model Based Reinforcement Learning for Atari." International Conference on Learning Representations. 2019.

---

> ### Author Response · Authors · 2022-11-17
> **Response to Reviewer 6X3a**
>
> We thank the reviewer for their insightful comments and their valuable time! In the following we would like to address these points:
>
> > It is not clear whether transformer-based model is indeed better than previous architecture with GRU of DreamerV2. From the current results, we cannot know whether the gain comes from additional techniques or transformer-based world models or additional computes. How does it compare to DreamerV2 trained with the proposed techniques, similar computes (not parameters), and hyperparameters to be more sample-efficient?
>
> Unfortunately, results for DreamerV2 on the Atari 100k benchmark are not available and producing them would be far beyond our computing capabilities (also see our comment on reviewer qSfN).
>
> > The paper says training is more efficient as it's parallel, but does not provide any measurement or investigation to explicitly show this compared to previous architecture based on a recurrent architecture. Gviven that the method utilizes a quite large model consisting of 10-layers Transformers, this should be further thoroughly supported.
>
> We provided wall-clock times for our world model. In the final revision we will add a comparison to previous methods.
>
> > It is written that leveraging Transformer-XL [Dai et al., 2019] can be good for long-term inference, but in the paper it's not investigated whether this is indeed the case. Additional results that compared with vanilla Transformer architecture could be included to support this.
>
> We made this design choice since the Transformer-XL architecture shows improved performance compared with vanilla Transformers (see Dai et al., 2019), because of the relative positional encodings, while also being faster. Our paper focuses on the aspects most relevant for reinforcement learning.
>
> > It is claimed that long-term dependencies can be captured with Transformers but final policy is trained on top of the representation from observation consisting of 4 stcaked frames. So it's difficult to know it's really good for capturing long-term dependencies in terms of either prediction for generating training samples or inference. It should be evaluated on more challenging tasks that require memories?
>
> We consider the Atari 100k benchmark to be a very challenging task in reinforcement learning and show excellent results. To better understand the role of the transformer we have included plots of its attention (see Section 3.2 and Appendix A.1) and an ablation study with less time steps into the past (see Appendix A.1).
>
> > For thresholded entropy loss, I can understand that it can make normalized entropy stable, but is this necessarily better than changing entropy throughout the training? How does the performance is related to the change in the entropy here?
>
> Our motivation behind the thresholded entropy loss is to ensure sufficient exploration, which is especially important in the low data regime, where each action is crucial. This means that during the entire course of training the entropy should be kept in a sensible range. Changing the entropy during training, e.g., linearly annealing the entropy coefficient, could prevent the entropy from diverging, but it could still easily collapse to zero. Our thresholded loss is an easy approach to this problem that does not require fine-tuning each game individually.
>
> > For sampling scheme, it would be nice to further describe and formulate how this cheme would affect the optimization objectives compared to using uniform sampling scheme. Moreover, ablation to see how this affects performance would be also nice.
>
> This is already shown in Figures 7 and 17 (Figure 5 in initial submission), which shows the performance for the balanced sampling scheme vs. the uniform sampling scheme.
>
> > Investigation into the effect of conditioning the policy on $z_t$ is not complete. It's okay to say that this would incur more costs, but it seems an important difference to prior approaches, so there should be an investigation into this with additional results and also corresponding wall times.
>
> We included an ablation study that conditions the policy on both $z_t$ and $h_t$, which shows that the score can get worse in this case. The policy is 3 times slower (see Appendix A.1, “Wall-Clock Times”).
>
> > Comparison with prior works can be much more informative if it includes some metrics in terms of training costs -- as it seems like the proposed method would require much more resources than other approaches (of course it's my guess because related information is not included anywhere, correct if i'm wrong), and it also does some pre-training on initial exploration samples.
>
> We are currently evaluating our method on a NVIDIA P100 GPU, so we can compare it with previous wall-clock times of SPR [1], Rainbow [2], and SimPLe [3], provided by [1]. We will add this comparison in the final revision.

---

> > ### Comment · Reviewer_6X3a · 2022-11-17
> > **Response**
> >
> > Thanks for your detailed comments! I'd like to put my response here on some points before the discussion period ends:
> >
> > | Unfortunately, results for DreamerV2 on the Atari 100k benchmark are not available and producing them would be far beyond our computing capabilities (also see our comment on reviewer qSfN).
> >
> > I understand that full comparison with DreamerV2 on Atari 100k would be too costly, but at least the paper could compare with publicly available DreamerV2 results on standard Atari benchmark instead of 100k benchmark on some of the tasks.
> >
> > | We provided wall-clock times for our world model. In the final revision we will add a comparison to previous methods.
> >
> > It's nice to include this, but without comparison with previous methods, it's difficult to see whether the proposed method is more efficient in which sense to supprort the claim made in the paper (Training is efficient ... (compared to RNN)).
> >
> > | We made this design choice since the Transformer-XL architecture ... Our paper focuses on the aspects most relevant for reinforcement learning.
> >
> > I still think it's important to explicitly show the importance of this design choice, as the paper claims that the proposed method has beneficial property 3 (Inference is efficient ... (compared with vanilla Transformers)) from adopting this design choice.
> >
> > | Our thresholded loss is an easy approach to this problem that does not require fine-tuning each game individually.
> >
> > Thanks for clarifying the intuition of introducing the thresholded entropy loss. But it seems like my question on the performance difference with and without the loss is not addressed yet as Figure 14 is only reporting the normalized entropies.
> >
> > | We consider the Atari 100k benchmark to be a very challenging task in reinforcement learning and show excellent results. To better understand the role of the transformer we have included plots of its attention (see Section 3.2 and Appendix A.1) and an ablation study with less time steps into the past (see Appendix A.1).
> >
> > Thanks for additional analysis. I agree with the authors in that Atari 100k could be challenging but it's questionable to me whether it's difficult due to the difficulty of capturing long-term dependency. And if so, whether baselines based on recurrent networks cannot handle this long-term dependency is still not clear.

---

> > > ### Author Response · Authors · 2022-11-20
> > > **Response**
> > >
> > > > It's nice to include this, but without comparison with previous methods, it's difficult to see whether the proposed method is more efficient in which sense to supprort the claim made in the paper (Training is efficient ... (compared to RNN)).
> > >
> > > We have now included a comparison to previous methods in terms of wall-clock time in Table 3.
> > >
> > > > I still think it's important to explicitly show the importance of this design choice, as the paper claims that the proposed method has beneficial property 3 (Inference is efficient ... (compared with vanilla Transformers)) from adopting this design choice.
> > >
> > > We have now included the total training time with a vanilla transformer, which is 1.5 times higher. We also show that the inference is almost twice as fast with Transformer-XL compared with a vanilla transformer.
> > >
> > > > Thanks for clarifying the intuition of introducing the thresholded entropy loss. But it seems like my question on the performance difference with and without the loss is not addressed yet as Figure 14 is only reporting the normalized entropies.
> > >
> > > We have now added the missing scores to Figure 15 (was Figure 14 previously), and added another game to the ablation study.
> > >
> > > > Thanks for additional analysis. I agree with the authors in that Atari 100k could be challenging but it's questionable to me whether it's difficult due to the difficulty of capturing long-term dependency. And if so, whether baselines based on recurrent networks cannot handle this long-term dependency is still not clear.
> > >
> > > Maybe the term *long-term dependency* is not clearly defined. In our view the 16 time steps that our world model considers are already long-term, compared with only one time step that a naive feedforward neural network would take into account (or e.g. 4 time steps). In that sense, long-term dependencies are important even for Atari, as shown in our analysis and ablation study. Recurrent neural networks are able to handle these dependencies as well, but they have to compress everything into the hidden state (where information can get lost) and cannot reconsider previous time steps, which is in contrast to transformers.

---

> > > > ### Comment · Reviewer_6X3a · 2022-11-22
> > > > **Response**
> > > >
> > > > Thank for your response! It's good to see that more claims made in the paper are supported with empirical evidences (Efficiency of Transformer-XL). I also have seen the reviews of the similar submission in ICLR, and it seems like I'm the only one who still has concerns -- thus I'm increasing the score to 5 and have no objection to accepting the paper. But I'm still leaving the notes here which I see as a remaining weakness of the paper in my point of view.
> > > >
> > > > - Comparison with RNNs
> > > >
> > > > What makes me difficult to recommend the paper to be accepted is that the paper is still not supporting its claim on the benefit of Transformers against the RNNs used in prior works. What would be the benefit of accessing information of previous time steps in practice for solving the tasks? Literature on video prediction shows that RNN can handle hundreds of time steps [Babaeizadeh et al., 2021]. Then, in which kind of RL tasks we can see the benefit of that empirically? Even if the paper is accepted, I'd like to strongly encourage the authors to consider this kind of question to support a claim on the benefit of leveraging Transformers.
> > > >
> > > > [Babaeizadeh et al., 2021] Babaeizadeh, Mohammad, Mohammad Taghi Saffar, Suraj Nair, Sergey Levine, Chelsea Finn, and Dumitru Erhan. "FitVid: Overfitting in pixel-level video prediction." arXiv preprint arXiv:2106.13195 (2021).
> > > >
> > > > - Tasks that require long-term dependency
> > > > New experiments show that learning the policy on top of 4 stacked frames almost does not make a big difference against learning the policy on top of Transformer features, which makes me difficult to see that this is really long-term. It might be worth to consider some navigation tasks (DMLab) or tasks introduced in [Chen et al., 2022]
> > > >
> > > > [Chen et al., 2022] Chen, Chang, Yi-Fu Wu, Jaesik Yoon, and Sungjin Ahn. "Transdreamer: Reinforcement learning with transformer world models." arXiv preprint arXiv:2202.09481 (2022).

---

### Official Review · Reviewer_gDHw · 2022-10-25

**Confidence:** 3
**Correctness:** 3
**Technical Novelty And Significance:** 3
**Empirical Novelty And Significance:** 3
**Recommendation:** 6

**Clarity, Quality, Novelty And Reproducibility:**

The paper is very clear although I found the experiments (and lack of theory) underwhelming in their depth of analysis for what I think is a very cool model. The main novelties here are how the authors designed the transformer. They introduce reward into the input and utilize a bunch of data and loss engineering. The results are reported with error bars and the standard for model adjudication in RL, and so I believe these are reproducable.

**Strength And Weaknesses:**

Strengths
- Their approach outperforms baselines across the board on Atari 100K.
- They train a lighter weight policy using the transformer world model, which makes inference faster than it would be otherwise.
- The method is well explained.
- The authors included code as their SI.

Weaknesses
- Missing a citation and discussion of Micheli et al., Transformers are Sample Efficient World Models. I do not believe this is published in a conference yet so we can consider it concurrent work but you should include it in your paper.
- There's no analysis of the world model! Tell me more about what the transformer has learned: is it a veridical representation of the games, are certain elements overemphasized over others? What's going on with the world model? How does its representations change when you add reward as an input?
- Results are only reported for one challenge (Atari 100K) and there's clearly a large amount of tweaking that went into this work in terms of loss and data engineering. That's understandable to an extent, but as it is paired with the limited insights into what is happening in the world model, I was left underwhelmed by this work.
- The title is flashy, but it also raises an important question which I don't believe was properly addressed in this paper: what is the sample efficiency of the transformer world model when you train on different amounts of data? For instance, try log-spaced intervals from 10 to the 100K reported in the main text. Understanding what failures occur at which dataset sizes would go a long way in illustrating the benefits (and persistent failure modes) of transformers as world models, and how introducing reward changes that sample efficiency curve.

Questions
- In your intro you write: "This can be illustrated by a simple example: in the game of Pong, the paddles and the ball move independently. A successfully trained world model is able to imagine trajectories with paddle and ball configurations that have never been observed before, which enables learning of improved behaviors." Is there evidence for this? Or are you positing that world models could help an agent solve this out-of-domain challenge?
- "In contrast to Hafner et al. (2021), the role of the observation model is to capture only non-temporal information about the current time step. However, we include short-time temporal information, since a single observation ot consists of four frames (aka frame stacking, see also Section 2.2)." I think the first sentence here is funky. Did you mean "in contrast to Hafner, *where* the role..."
- "Due to these design choices, our world model combines the benefits of transformers and recurrent neural networks..." I'm not sure any of the four bullets are areas where RNNs are better than transformers. My thought is the key benefit of RNNs over transformers is in the amount of memory needed to run them (i.e., RNNs operate on a single token at a time whereas transformers look at many at once).

**Summary Of The Paper:**

The authors investigate the effectiveness of transformers as world models. Their main innovation is to input not just states and actions (as was done in earlier works like Ha & Schmidhuber, 2018), but also to add rewards (predicted or real). This additional constraint on the input interacts well with the transformer architecture to yield sample efficient and performant agents on the Atari 100k benchmark.

UPDATE

raising my score due to the reviewers responses and work over the rebuttal period

**Summary Of The Review:**

I was really excited about this paper for the possibility of understanding why transformers might make sample efficient world models. Instead, this paper is a standard benchmark paper describing a recipe for performant transformer-based world models. I am far less interested in the latter, but I can appreciate that the contribution required a lot of work and is nontrivial. So I'm borderline on this paper. It would make a fantastic workshop paper. I'm less sure if ICLR is the right venue. Looking forward to seeing what the other reviewers think.

---

> ### Author Response · Authors · 2022-11-17
> **Response to Reviewer gDHw**
>
> We would like to thank the reviewer for their time and the detailed review.
>
> > Missing a citation and discussion of Micheli et al., Transformers are Sample Efficient World Models. I do not believe this is published in a conference yet so we can consider it concurrent work but you should include it in your paper.
>
> In the revision we cite the arXiv submission of Micheli et al. [1], which was developed independently of our work and shows a very similar approach and similar performance.
>
> > There's no analysis of the world model! Tell me more about what the transformer has learned: is it a veridical representation of the games, are certain elements overemphasized over others? What's going on with the world model? How does its representations change when you add reward as an input?
>
> In our revised version we added analysis of our world model, including imagined trajectories and insights into the attention of the transformer (see Section 3.2 and Appendix A.1).
>
> > Results are only reported for one challenge (Atari 100K) and there's clearly a large amount of tweaking that went into this work in terms of loss and data engineering. That's understandable to an extent, but as it is paired with the limited insights into what is happening in the world model, I was left underwhelmed by this work.
>
> We are quite excited that (concurrently with another ICLR 2023 submission, see general comment) we could show that transformers are a good match for the sample-efficient scenario and show excellent performance on the Atari 100k benchmark. Of course more benchmarks would be interesting, however, we think that the Atari 100k benchmark is versatile and contains many different kinds of games. See for example [2, 3, 4], who also consider solely the Atari benchmark.
>
> > The title is flashy, but it also raises an important question which I don't believe was properly addressed in this paper: what is the sample efficiency of the transformer world model when you train on different amounts of data? For instance, try log-spaced intervals from 10 to the 100K reported in the main text. Understanding what failures occur at which dataset sizes would go a long way in illustrating the benefits (and persistent failure modes) of transformers as world models, and how introducing reward changes that sample efficiency curve.
>
> That is a great suggestion. We are currently running experiments and expect results before the end of the discussion period.
>
> > In your intro you write: "This can be illustrated by a simple example: in the game of Pong, the paddles and the ball move independently. A successfully trained world model is able to imagine trajectories with paddle and ball configurations that have never been observed before, which enables learning of improved behaviors." Is there evidence for this? Or are you positing that world models could help an agent solve this out-of-domain challenge?
>
> Generating data that has not been in the training sequences is one of the main motivations of world models, and we believe that it is the main reason for its good performance. However, it is difficult to check whether the generated trajectories do exist in the training data. We reformulated the passage for clarification.
>
> > "In contrast to Hafner et al. (2021), the role of the observation model is to capture only non-temporal information about the current time step. However, we include short-time temporal information, since a single observation $o_t$ consists of four frames (aka frame stacking, see also Section 2.2)." I think the first sentence here is funky. Did you mean "in contrast to Hafner, where the role..."
>
> We simplified the first sentence to improve clarity of the statement.
>
> > "Due to these design choices, our world model combines the benefits of transformers and recurrent neural networks..." I'm not sure any of the four bullets are areas where RNNs are better than transformers. My thought is the key benefit of RNNs over transformers is in the amount of memory needed to run them (i.e., RNNs operate on a single token at a time whereas transformers look at many at once).
>
> You are right, we modified that part.

---

> > ### Author Response · Authors · 2022-11-17
> > **Response to Reviewer gDHw (continued)**
> >
> > > The paper is very clear although I found the experiments (and lack of theory) underwhelming in their depth of analysis for what I think is a very cool model.
> >
> > We hope that our revised version (including additional ablation experiments, clarifications, and an analysis of the world model) sparks your excitement about our submission again.
> >
> > **References:**
> > [1] Micheli, Vincent, Eloi Alonso, and François Fleuret. "Transformers are sample efficient world models." arXiv preprint arXiv:2209.00588 (2022).
> > [2] Schwarzer, Max, et al. "Data-Efficient Reinforcement Learning with Self-Predictive Representations." International Conference on Learning Representations. 2020.
> > [3] Badia, Adrià Puigdomènech, et al. "Agent57: Outperforming the atari human benchmark." International Conference on Machine Learning. PMLR, 2020.
> > [4] Hessel, Matteo, et al. "Rainbow: Combining improvements in deep reinforcement learning." Thirty-second AAAI conference on artificial intelligence. 2018.

---

> > ### Comment · Reviewer_gDHw · 2022-11-17
> > **Response**
> >
> > > In the revision we cite the arXiv submission
> >
> > I didn't realize this was just an arXiv. I had seen this months ago but your response makes sense. Thanks.
> >
> > > In our revised version we added analysis of our world model, including imagined trajectories and insights into the attention
> >
> > This is fantastic, thanks!
> >
> > > ...however, we think that the Atari 100k benchmark is versatile and contains many different kinds of games...
> >
> > Looking at my comment again, I was probably being unfair. I don't find Atari that convincing but it is the standard in the field. Looking forward to extensions of your work on other benchmarks.
> >
> > > That is a great suggestion. We are currently running experiments...
> >
> > Looking forward to these! I'm still on the fence (although much more positive than before given the new results) but some computational results, like sample efficiency, would significantly improve this paper and bring it above threshold IMO.

---

> > > ### Author Response · Authors · 2022-11-19
> > > **Response**
> > >
> > > > Looking forward to these! I'm still on the fence (although much more positive than before given the new results) but some computational results, like sample efficiency, would significantly improve this paper and bring it above threshold IMO.
> > >
> > > We just uploaded a new revision of our paper that now contains your requested experiments: Table 2 shows results of our TWM after 5K, 10K, 25K, 50K, 75K, and 100K interactions. TWM outperforms SimPLe after only 25K interactions (in terms of mean normalized score) and achieves better results than SPR after 50K interactions. We also added an ablation study where we train some games again with a lower number of interactions but the same training budget. We hope that these results convince you and we would be happy if you would reconsider your score.

---

### Official Review · Reviewer_qSfN · 2022-10-25

**Confidence:** 3
**Correctness:** 3
**Technical Novelty And Significance:** 2
**Empirical Novelty And Significance:** 3
**Recommendation:** 6

**Clarity, Quality, Novelty And Reproducibility:**

The paper is well-written generally, providing clear reasoning throughout the whole manuscript. Results might be reproduced with reasonable effort.

**Strength And Weaknesses:**

### Strength
- The paper is well-written generally, providing clear reasoning throughout the whole manuscript.
- Experiments on the Atari 100K benchmark showed an overall improvement over several model-free and model-based methods.
- Detailed ablation studies have been provided.

### Weakness

- Major:
    - Is the input for policy learning $z_t$ or the Transformer predicted $\hat{z}_t$ during policy learning? The description in the `Choice of Policy Input` section looks confusing. If it's $\hat{z}_t$, did you apply the frame stacking?
    - It seems this work has been inspired by the Dreamer framework a lot, why didn't the author choose to compare it with Dreamer-V2?
    - The results on the effect of taking rewards as the input seems too weak, with benefits only shown on one task. Could the authors please explain how are the tasks selected in the ablation studies?
    - Since Atari games do not require long-term memory in general, how does the Transformer help on this Atari 100K benchmark?

- Minor:
    - The objectives seem complex, could the authors analyze the sensitivity of the set of hyper-parameters used in the objectives?
    - How are the M trajectories generated given N sequences used to train the world model?

**Summary Of The Paper:**

This paper proposed a transformer-based world model. Utilizing the Transformer-XL, the agent is trained in imagination rollouts in a computationally efficient way. Though inspired by prior work, such as the Dreamer frameworks, several modifications were made and ablation studies showed benefits to the final performance. Experiments on the Atari 100K benchmark showed an overall improvement over a set of the prior model-free and model-based methods.

**Summary Of The Review:**

This paper proposed a transformer-based world model. Experiments on the Atari 100K benchmark showed an overall improvement over a set of the prior model-free and model-based methods. However, the motivation is not clear and some key statements need further explanations.

---

> ### Author Response · Authors · 2022-11-17
> **Response to Reviewer qSfN**
>
> We thank the reviewer for their detailed review.
>
> > Is the input for policy learning $z_t$ or the Transformer predicted $\hat{z}_t$ during policy learning? The description in the `Choice of Policy Input` section looks confusing. If it's $\hat{z}_t$, did you apply the frame stacking?
>
> During policy learning the input to the policy is $\hat{z}_t$, the latent state produced by the world model. Since it is in latent space, there is no frame stacking at this point. We clarified this in the "Choice of Policy Input" section.
>
> > It seems this work has been inspired by the Dreamer framework a lot, why didn't the author choose to compare it with Dreamer-V2?
>
> A comparison against DreamerV2 on the Atari 100k benchmark would be very interesting. However, until now those results are not available and reproducing them is beyond the compute capacity of our group. We agree that the RL community would highly benefit from this evaluation. Note that for similar reasons the comparison against DreamerV2 is missing in other popular papers, e.g., Ye et al. (2021), Micheli et al. (2022).
>
> > The results on the effect of taking rewards as the input seems too weak, with benefits only shown on one task. Could the authors please explain how are the tasks selected in the ablation studies?
>
> The reported games are selected randomly. In our revision we expand the ablation study to more games and observe that there also is a positive effect (see Figure 8).
>
> > Since Atari games do not require long-term memory in general, how does the Transformer help on this Atari 100K benchmark?
>
> The transformer can directly access previous states simultaneously, instead of a compressed state of an RNN, and can model more complex dependencies between them. It also allows us to easily integrate other modalities (in our case actions and rewards) of past time steps (also see Section 2.1, “Autoregressive Dynamics Model”). In our revision we added attention maps that demonstrate that the transformer really makes use of past information. We also did an ablation study that shows that reducing the number of time steps in the transformer worsens the performance (see Appendix A.1).
>
> > How are the M trajectories generated given N sequences used to train the world model?
>
> We select $M$ of the $N \times \ell$ observations and encode them into latent states to serve as initial states for new trajectories. The dynamics model iteratively generates these $M$ trajectories based on actions provided by the policy. We clarified this in Section 2.3.
>
> > This paper proposed a transformer-based world model. Experiments on the Atari 100K benchmark showed an overall improvement over a set of the prior model-free and model-based methods. However, the motivation is not clear and some key statements need further explanations.
>
> We hope that our answers could provide more motivation and clarify some of the details of our paper! Please let us know more questions you might have.
>
> **References**:
> Ye, Weirui, et al. "Mastering atari games with limited data." Advances in Neural Information Processing Systems 34 (2021): 25476-25488.
> Micheli, Vincent, Eloi Alonso, and François Fleuret. "Transformers are sample efficient world models." arXiv preprint arXiv:2209.00588 (2022).

---

### Author Response · Authors · 2022-11-17
**Revised version and concurrent ICLR submission**

**Changes in the revised version**:
1. Added analysis of the world model:
    - Trajectories imagined by the world model showing different scenarios, including stochasticity of the world model and a very long trajectory
    - Plots of attention maps of the transformer
    - We provide information on the wall-clock times of our method. We will add a comparison with previous approaches in the final revision.
2. Added two new ablation studies:
    - We reduce the number of time steps for the transformer and show that this can lead to lower scores.
    - We condition the policy on $[z,h]$ and show that this can lower the final score.
3. Extended the ablation study “No Rewards” to 8 games, and “Uniform Sampling” to 4 games
4. Clarified the mentioned passages in the text
(New in second revision:)
5. Added information on sample efficiency:
    - We provide scores of our main experiments after different amounts of interactions (5K, 10K, 25K, 50K, 75K, 100K), which show that the normalized score of our method already exceeds previous methods after 50K interactions and SimPLe after 25K interactions.
    - Added an ablation study on a subset of games to find out whether the sample efficiency can be further improved. We train on these games again with a smaller amount of interactions (10K, 25K, 50K) but the same training budget as for 100K.
6. Improved the “Thresholded Entropy Loss” ablation study by providing the human normalized scores.
7. Extended wall-clock times section:
    - Added comparison to previous sample-efficient methods
    - Added training time for vanilla transformer
    - Added throughput for vanilla transformer
8. Added visualizations of frame stacking
(New in third revision:)
9. Added another game to the "Thresholded Entropy Loss" ablation study
10. Improved appearance of figures

**Concurrent ICLR submission**:
Note that we became aware of a concurrent ICLR 2023 submission under https://openreview.net/forum?id=vhFu1Acb0xb, which also considers transformer-based world models for the 100k Atari benchmark. The authors also present similarly strong results and a similar approach, which was well received. We would appreciate it if it is possible that both submissions were looked at by one person (ideally on the area chair level) to ensure consistent scores and fair evaluation.

---

### Comment · Area_Chair_ZGur · 2022-11-18
**Responses**

Dear Reviewers,

Thank you for your discussion. Do you have further comments for the authors? Have you changed your score do to authors/other reviewers response?

Kind Regards,
AC

---

### Comment · Area_Chair_ZGur · 2022-11-22
**Questions**

Dear Reviewers,

Reviewer qSfN - Have author's responses satisfied you - have you changed your opinion about this score?
Reviewer gDHw - You seem more satisfied with the paper after author's replies. What your current opinion for the score of the paper?

The authors point out that there is another quite similar paper: https://openreview.net/forum?id=vhFu1Acb0xb
It would be very helpful if you had time to look at it and see if there are major differences as it is quite similar but has much higher scores.

Thank you,
AC

---

> ### Comment · Reviewer_qSfN · 2022-11-23
> **Response**
>
> Dear AC,
>
> Two of my main concerns are still not addressed by the authors. I choose to stay with my original score.
> 1) I asked the author to compare with DreamerV2:
>     The authors pointed to two other works, EfficientZero, Ye et al. (2021) and one concurrent work, Micheli et al. (2022), that also lack the comparison.
>
> EfficientZero, building upon MuZero, improved the data efficiency significantly over MuZero. For the concurrent work, a comparison with DreamerV2 was also proposed by reviewers. A lesson I learned from the concurrent work is that agent learning can be benefitted from accurate and consistent generation. The concurrent work does not use transformer output as the state representation for policy learning. Instead, they decode $z_t$ back to image space $\hat{x}_t$, and learn $\pi(a_t | \hat{x}_t)$. They utilized VQVAE to learn latent state embedding.
> Without the comparison, it's not clear whether Transformer-based model-based RL is better than RNN-based model-based RL. Indeed, under model-free RL setting, the Transformer-based model does not outperform the RNN-based model on reactive tasks, Parisotto et al. (2020). So results of the comparison with DreamerV2 are important to share with the community for a better understanding of the advantages of Transformer-based model-based RL.
>
> 2) Why does long-term dependency benefit agent learning?
>     The author provided the attention map and additional ablation study on varied history lengths. It shows, with a shorter history length, the performance degraded.
>
> This experiment only shows that the Transformer-based model requires a longer context sequence to learn a good state representation. Again, I think a comparison with the RNN-based model on memory-required tasks is necessary to address this concern.
>
> Thus, based on the current version, the lesson I get from this work is that, by utilizing a Transformer-based world model, the agent may learn faster at the early training stage given a limited interaction buffer(, and might converge to a suboptimal). However, this is not what the author tried to deliver, long-term dependency improves sample efficiency.
>
> References:
> Ye, Weirui, et al. "Mastering atari games with limited data." Advances in Neural Information Processing Systems 34 (2021): 25476-25488.
> Micheli, Vincent, Eloi Alonso, and François Fleuret. "Transformers are sample efficient world models." arXiv preprint arXiv:2209.00588 (2022).
> Parisotto, Emilio, et al. "Stabilizing transformers for reinforcement learning." _International conference on machine learning_. PMLR, 2020

---

### Author Response · Authors · 2022-12-07
**Two quite similar papers with very different scores...**

Dear (senior) area chair,
dear reviewers,

thank you again for considering our paper for ICLR.

As we pointed out, the other quite similar paper at  https://openreview.net/forum?id=vhFu1Acb0xb that similar to us propose transformer-based world models received the scores of 8, 6, 8, 8.  Their scope, results and experiments are very similar to ours.  Currently we have 5, 6, 5, 8.

Could you point out to us how their paper is significantly different to ours?

Best regards,
the authors

---

### Decision · Program_Chairs · 2023-01-20

**Decision:**

Accept: poster

**Justification For Why Not Higher Score:**

No comparison to LSTM/DreamerV2 on the same benchmark (Atari 100k) or longer training.

**Justification For Why Not Lower Score:**

Good performance, nice to compute policy from codes.

**Metareview: Summary, Strengths And Weaknesses:**

The paper trains a world model on Atari and trains policy on the model itself only, to achieve good performance on Atari 100k. The novelty of this paper (the setup itself existed before) is to use transformer for the world model: Consuming all the vector quantised codes of a given frame at the time, updating the transformer, sampling next set of codes. The policy is computed from the codes only.
- The positive of this paper is showing that this works well beating previous methods published on this benchmark.
- The negative is that the setup is not novel - a similar system has been trained on with LSTM instead of transformer (DreamerV2) but not evaluated in the same setting (100k interactions). One of the biggest drawbacks if the paper is that the authors did not rerun the same system with LSTM instead of transformer or compared to that paper in the same setting.

**Note From Pc:**

if the above contains the word "oral" or "spotlight" please see: "oral" presentation means -> notable-top-5% and "spotlight" means -> notable-top-25%. As stated in our emails, we are disassociating presentation type from AC recommendations

**Summary Of Ac-Reviewer Meeting:**

We didn't have an in person meeting, but had a lot of exchanges and in the end all the reviewers agreed on accept (so perhaps it is not borderline anymore).